



# Enhancing dust aerosols monitoring capabilities across North Africa and the Middle East using the A-Train satellite constellation

Anna Moustaka[1,2,3], Stelios Kazadzis[2], Emmanouil Proestakis[3], Anton Lopatin[4], Oleg Dubovik[5], Kleareti Tourpali[1], Christos Zerefos[6,7,8,9], Vassilis Amiridis[3], and Antonis Gkikas[3,6]

[1] Department of Physics, Aristotle University of Thessaloniki, 54124, Greece
[2] Physicalisch Meteorologisches Observatorium, World Radiation Center, Davos, 7260, Switzerland
[3] Institute for Astronomy, Astrophysics, Space Applications and Remote Sensing, National Observatory of Athens, Athens, 11810, Greece
[4] GRASP SAS, Lille, 59000, France
[5] Laboratoire d'Optique Atmosphérique, CNRS/Université de Lille, Villeneuve-d'Ascq, 59650, France
[6] Research Centre for Atmospheric Physics and Climatology, Academy of Athens, Athens, 11521, Greece
[7] Biomedical Research Foundation, Academy of Athens, Athens, 11527, Greece
[8] Navarino Environmental Observatory (N.E.O.), Messinia, 24001, Greece
[9] Mariolopoulos-Kanaginis Foundation for the Environmental Sciences, Athens, 11251, Greece

*Correspondence to*: Anna Moustaka (anna.moustaka@pmodwrc.ch)

**Abstract.** North Africa and the Middle East encompass the most active dust sources on the planet. Due to the limited availability of ground-based aerosol observations across the deserts, spaceborne retrievals represent the most reliable source of information for monitoring dust particles over these vast areas. In the current study, we present a synergistic approach incorporating aerosol retrievals acquired by active (CALIOP) and passive (POLDER-3, MODIS) instruments mounted on satellites of the A-Train constellation. Our main objective is to dynamically (in terms of space and time) estimate the dust lidar ratio (LR) throughout the CALIPSO operational period (2006-2023) by collocating columnar aerosol optical depth observations (POLDER-3/GRASP, MIDAS) and vertically resolved dust aerosol profiles obtained by CALIPSO. According to our findings, the agreement among the satellite-based retrievals improves, when the default and constant CALIPSO dust LR (44 sr) is adjusted. Specifically, increasing and decreasing dust LR tendencies are recorded in North Africa and in the Middle East, respectively, whereas a narrow transition zone of neutral declinations appears between the two regions. Furthermore, compared to the standard dust LR, higher values are recorded east of the Caspian Sea, while negative departures are found in Iran and southern Pakistan. The proposed dust LR adjustments are maximized over/near dust sources and during the dry period of the year, indicating a dependence on dust activity. The evaluation against AERONET observations clearly demonstrates that the refined dust LRs improve CALIOP's performance when mineral particles are probed, but further justification is needed. The integration of the refined dust LRs significantly modifies the standard CALIPSO dust optical depth (DOD) climatological patterns, particularly in the Western Sahara, the Bodélé depression (northern Chad Basin), the Libyan Desert and Saudi Arabia. Positive and negative shifts in the regional DOD timeseries are found in North Africa and in the Middle East, respectively, with no notable modifications on the inter-annual and intra-annual trends in either region. A key priority of the analysis carried out is to improve the efficiency of CALIPSO dust retrievals towards advancing their utility in a wide range of scientific



applications. The advent of the EarthCARE satellite mission, along with the incorporation of new aerosol models into the forthcoming CALIPSO Version 5 aerosol retrieval algorithm, will serve as a reference for our calculations. In this context, our findings also highlight that a synergy of multisensor aerosol products with modelling tools can enhance the spatiotemporal

representation of aerosol properties, such as the lidar ratio, further improving retrieval utility.

## 1 Introduction

Dust aerosols represent a significant component of the global aerosol burden, profoundly influencing climate and various environmental processes (Zender et al., 2004; Textor et al. 2006; Mahowald et al., 2014; Gkikas et al., 2021; Logothetis et al., 2021; Gkikas et al; 2022; González-Romero et al., 2023; Castellanos et al., 2024). The most prominent dust sources are located

in arid and semi-arid regions, where the prevailing meteorological conditions and the soil characteristics facilitate the mobilization of dust particles (Ginoux et al., 2012; Prospero et al., 2013; Huang et al., 2014; Chen et al., 2018). North Africa features some of the most active dust sources worldwide. The Bodélé Depression in the northern Chad Basin, is recognized as the most prolific single source of mineral dust globally (Warren et al., 2007; Ginoux et al. 2012; Kok et al., 2021), with substantial emissions also emanating from the western Sahara, the eastern Libyan Desert, and the Nubian Desert (Sudan and

Egypt) (Engelstaedter et al. 2006). The spatial extent and intensity of these sources vary significantly depending on the prevailing winds and surface conditions (Mbourou et al., 1997; Schepanski, 2018; Tindan et al., 2023).

The Arabian Peninsula also serves as a major source region for mineral dust aerosols. The Rub' al Khali desert, one of the largest continuous sand seas in the world, contributes substantially to regional and global dust loading (Jish Prakash et al. 2015). Other significant sources include alluvial plains within the Tigris-Euphrates river system and sandy deserts across the

region (Pease et al., 1998; Hamidi et al., 2013). Dust emissions across the Arabian Peninsula show strong seasonality, influenced by the prevailing winds and the availability of loose, erodible material on the surface (Yu et al., 2013; Yu et al., 2015; Abdul-Wahab et al. 2017; Solmon et al. 2015). The transport of dust from this region significantly affects the atmospheric composition and radiative balance of the Middle East and western parts of Asia (Jish Prakash et al., 2015). Desert regions in Central Asia contribute to global dust emissions, albeit typically at lower intensities than those in North Africa and

the Arabian Peninsula (Xiong et al., 2020). Among these, the Karakum Desert (Turkmenistan) and the Kyzylkum Desert (spanning Uzbekistan and Kazakhstan) stand out as smaller yet significant sources of regional dust loading (Li and Sokolik, 2018). These deserts share similar arid characteristics, but their dust production and transport are influenced by unique geographical factors and meteorological conditions (Banks et al., 2022).

The complex nature of mineral dust aerosols, in terms of particle sizes, shapes, and compositions (Castellanos et al., 2024),

leads to significant challenges for accurate remote sensing and modeling, necessitating the deployment of advanced observational techniques. Ground-based measurements employing passive and active remote sensing techniques have provided invaluable long-term data on aerosol intensive and extensive properties (Benkhalifa et al., 2017; Ningombam et al. 2019; Raptis et al. 2020; Yu et al., 2022; Zhang et al., 2022; Eom et al., 2022). Spaceborne observations from passive sensors, such



as those from the Moderate Resolution Imaging Spectroradiometer (MODIS) onboard Terra and Aqua satellites, have delivered

multi-year records of columnar aerosol optical depth (AOD) with near-global coverage since 2000 (Luo et al., 2014; Mao et al., 2014; Gupta et al., 2020; Gkikas et al., 2021; Kang et al., 2023). The recently developed MODIS Dust Aerosol (MIDAS) dataset (Gkikas et al., 2021; 2022) further refines dust-specific AOD by combining MODIS-Aqua quality-assured AOD retrievals with Modern-Era Retrospective analysis for Research and Applications version 2 (MERRA-2; Gelaro et al., 2017) reanalysis dust fraction ratios (in optical terms), offering a high-resolution record of daily dust optical depth (DOD) on a global

scale, spanning from 2003 to 2017 (Gkikas et al., 2021). Additionally, the multi-angle, multi-spectral polarimetric (MAP) measurements have been providing comprehensive data on aerosol particle size, shape, and refractive index, enabling detailed characterization of aerosols' properties and interactions within the atmosphere since the mid-1990s (Hansen et al., 1995; Mishchenko et al., 1997). Polarimetric measurements from the Polarization and Directionality of the Earth's Reflectances (POLDER-3) on the PARASOL platform exemplify the contributions of MAP technology to aerosol science. When combined

with the Generalized Retrieval of Atmosphere (GRASP), the linearly polarized and total radiances from POLDER-3 sensor enables the retrieval of detailed aerosol properties, such as particle size distribution, complex refractive indices, and sphericity. Furthermore, POLDER-3/GRASP achieves high-quality aerosol optical depth (AOD) and absorption retrievals (Dubovik et al., 2011; Li et al., 2019; Chen et al., 2020; Zhang et al., 2021), validated effectively against AERONET stations (Chen et al., 2020; Zhang et al., 2021).

Along with space-borne passive sensors measuring the reflectance at the top of the atmosphere (TOA), active remote sensing techniques, offer superior capabilities for resolving the vertical distribution of aerosols throughout the atmosphere. From June 2006 to August 2023, the Cloud-Aerosol Lidar with Orthogonal Polarization (CALIOP) instrument onboard the Cloud-Aerosol Lidar and Infrared Pathfinder Satellite Observation (CALIPSO) satellite provided vertically resolved aerosol retrievals, both day and night, at high spatial and temporal resolutions, regardless the underlying surface type (Winker et al., 2010). Among

the prediscussed limitations of the CALIOP's elastic lidar in a number of previous studies (Winker et al., 2009; Ma et al., 2013; Kim et al., 2018; Gui et al., 2022; Haarig et al., 2018; Li et al., 2022; Moustaka et al., 2024), the accuracy of CALIPSO's AOD retrievals is highly contingent on the predetermined extinction-to-backscatter ratio, a key parameter defined as lidar ratio (LR), which varies significantly depending on aerosol types. This dependency can lead to inaccuracies, especially if the LR assigned to an aerosol type does not adequately reflect its true microphysical characteristics. Past studies indicated that

CALIPSO's AOD is often underestimated by approximately 13% when compared to AERONET. This underestimation is attributed to both aerosol misclassification and inaccuracies in the modeled microphysics for certain aerosol types, such as polluted dust or smoke (Schuster et al., 2012). Additionally, CALIOP's ability to measure particle linear depolarization at 532 nm has provided great insights into identifying non-spherical mineral dust particles (Burton et al., 2013).

Specifically, for the non-spherical dust particles, the study from Amiridis et al. (2013) has shown that the increase of dust LR

over the Sahara leads to better accuracy of AOD calculations by reducing the observed discrepancies compared to MODIS and AERONET. According to a more recent study from Moustaka et al. (2024), the authors used the DeLiAn database (Depolarization ratio, Lidar ratio, and Ångström exponent database; Floutsi et al., 2023), a collection of state-of-the-art ground-



based lidar observations acquired in areas characterized by different aerosol conditions, in order to estimate the effect of an

updated aerosol-speciated LR on AOD and subsequently to the direct radiative effects (DREs) across the North Africa, Middle

East and Europe domain (NAMEE). The use of DeLiAn-derived LRs improved accuracy, particularly for dust-dominated

regions, resulting in more reliable DRE calculations and specifically decreased discrepancies between CALIPSO and

AERONET observations. This notably enhanced the surface cooling and atmospheric warming estimates by up to 35% under

moderate-to-high aerosol loads. These findings along with those of past relevant studies (Omar et al., 2013; Konsta et al.,

2018) underscore the importance of accurately determining the dust LR, across major desert regions over the globe. The

adaptation of a more "realistic" dust LR, along with more reliable elastic lidar-derived AOD values, could contribute to a more

robust aerosol typing. This can be achieved through synergistic approaches by also implementing measurements from the

recently launched EarthCARE (Cloud, Aerosol, and Radiation Explorer) platform, which is a joint mission of the European

Space Agency (ESA) and the Japan Aerospace Exploration Agency (JAXA) (Illingworth et al., 2015; Wandinger et al., 2023).

In addition to the LR, other properties of dust, including its emission processess, mineralogical composition, and morphology,

are crucial for assessing its climatic role, as they influence key optical characteristics like DOD, single scattering albedo (SSA),

and refractive index, which are essential inputs for radiative transfer models (RTMs). A deeper understanding of these

properties allows for a more accurate representation of dust's microphysical characteristics and radiative effects, ultimately

improving climate predictions and reducing uncertainties in dust-related direct radiative effects (DREs) (Kok et al., 2017; Kok

et al., 2023).

Our study focuses on improving dust aerosol monitoring across North Africa and the Middle East by refining dust LR estimates

via a synergistic approach between active and passive sensors of the A-Train satellite constellation. Based on the CALIPSO

aerosol classification scheme, which uses accurate depolarization measurements of the CALIOP lidar to detect non-spherical

dust layers, we focus on dust cases. We aim to adjust the dust LR so that the CALIOP DOD at 532 nm matches the coincident

columnar observations from the MIDAS (from 2007 to 2017) and POLDER-3/GRASP (from 2006 to 2009) datasets (Section

2). The identification of the dust cases, along with the methodology for the adjustment of dust LRs between the different

sensors are provided in Section 3. In Section 4, we present the geographical distribution of dust LR on an annual and seasonal

basis. This analysis begins with the examination of the DOD retrievals used to adjust LR within the A-Train constellation.

Additionally, to evaluate the effect of dust LR on CALIOP's columnar DOD retrievals an assessment analysis using

AERONET ground-based observations has been performed. Following the evaluation analysis, we examine the impact of the

updated LRs on dust climatological patterns and regional levels. As a concluding step in assessing the adjusted dust LRs, we

present the inter-annual and intra-annual DOD time series for the 2007–2022 period, analyzed separately for North Africa and

the Middle East. Finally, Section 5 summarizes the main findings and discusses future aspects.



## 2 Data sets

### 2.1 CALIOP-CALIPSO Spaceborne Retrievals

From April 2006 to August 2023, the Cloud-Aerosol Lidar with Orthogonal Polarization (CALIOP) on the Cloud-Aerosol
Lidar and Infrared Pathfinder Satellite Observations (CALIPSO) satellite has significantly enhanced our understanding of
aerosol and cloud interactions and their roles in the climate system (Winker et al., 2009). CALIOP operates as a dual-
wavelength backscatter (532 and 1064 nm) and single-wavelength polarization (532 nm) lidar system with horizontal
averaging and vertical resolution variable based on both wavelength and altitude (Winker et al., 2009). This system aids in

distinguishing aerosols from clouds and differentiating between fine and coarse-mode aerosol layers, while its depolarization
signal is essential for distinguishing between spherical particles (e.g., marine particles, liquid cloud droplets) and non-spherical
particles (e.g., dust, ice cloud particles) within the atmosphere (McGill et al., 2007; Hu et al., 2009; Omar et al., 2009; Burton
et al., 2012; Burton et al., 2013; Kim et al., 2018).

The latest version 4.5 (V4.5) of the CALIOP level 2 (L2) aerosol classification product, besides the more comprehensive and

accurate definitions of stratospheric particles (Tackett et al., 2023), preserves the same rationality with the previous V4 versions
for the tropospheric aerosol layers, using layer-integrated values of depolarization and attenuated backscatter from L1 data,
along with geographical location, surface type, and layer altitude information, to assign distinct aerosol types (Kim et al.,
2018). The CALIPSO algorithm classifies tropospheric aerosols into seven categories: "marine," "dust," "polluted
continental/smoke," "clean continental," "polluted dust," "elevated smoke," and "dusty marine." Dust is one of the most well-

characterized subtypes (Burton et al., 2013), thanks to CALIOP's accurate depolarization measurements, which enable the
detection of non-spherical dust layers. Critical quality filters are applied during CALIOP-CALIPSO data processing to
minimize errors in layer detection, classification, and extinction retrieval, as well as biases from negative signal anomalies.
Cloud contamination is also mitigated using the cloud aerosol discrimination (CAD) algorithm and by applying misclassified
cirrus fringe filters (Tackett et al., 2018; Proestakis et al., 2018; Marinou et al., 2017).

In the present study, based on the CALIPSO aerosol subtype product, we identify dust scenes during daytime satellite orbits
and we exploit quality-assured (QA) vertical profiles of the backscatter coefficient at 532 nm, sourced from the LIVAS (Lidar
climatology of Vertical Aerosol Structure) database (Amiridis et al., 2015), for the computation of the columnar dust optical
depth (DOD), over a 17-year period (June 2006 to August 2023). LIVAS is a comprehensive 3-D multi-wavelength database
that provides global aerosol and cloud optical properties based on CALIPSO observations at 532 and 1064 nm with a resolution

of 1ºx1º. For the LIVAS development, the ESA-CALIPSO dataset (Wandiger et al., 2011), the EARLINET (The EARLINET
publishing group 2000–2010) and AERONET products (Holben et al., 1998; 2001), along with aerosol models in the literature
(Deshler et al., 1993; Wandiger et al., 1995; Omar et al., 2005; Sayer et al., 2012), were integrated. Since its inception, LIVAS
has supported numerous studies related to dust climatology and retrieval optimization (Marinou et al., 2017; Proestakis et al.,
2018; Moustaka et al., 2023; Moustaka et al. 2024), new dataset development (Fountoulakis et al., 2021; Fountoulakis et al.,



2022; Papachristopoulou et al., 2022), and the evaluation of aerosol effects on solar radiation models (Konsta et al., 2018; Moustaka et al., 2024).

## 2.2 ModIs Dust AeroSol (MIDAS) dataset

The ModIs Dust AeroSol (MIDAS) dataset (Gkikas et al., 2021) was developed to focus specifically on dust aerosols by estimating dust optical depth (DOD) at a global scale and fine spatial resolution (0.1° x 0.1°). The MIDAS product spans a 15-

year period (2003–2017) and integrates MODIS-Aqua AOD retrievals with the dust-specific information from the Modern-Era Retrospective analysis for Research and Applications version 2 (MERRA-2) reanalysis, which provides the dust fraction to the total aerosol load. The MIDAS development process includes multiple steps to ensure high-quality dust aerosol data, such as quality filtering, spatial and temporal colocation, and error estimation.

To generate the MIDAS DOD product, a synergy between MODIS-Aqua L2 AOD retrievals and MERRA-2 dust-to-total AOD

ratios are employed. The integration of MODIS AOD with MERRA-2 dust fraction requires careful spatial and temporal colocation. Since MERRA-2 outputs are available at a coarser spatial resolution (0.5° x 0.625°) compared to MODIS's fine-resolution grid (10 km x 10 km), the nearest MERRA-2 grid points to the MODIS swath-level retrievals are selected. The time colocation also considers the nearest hourly MERRA-2 data corresponding to the MODIS overpass. The MIDAS product employs a series of filtering criteria to enhance the accuracy of its AOD retrievals from MODIS. First, only high-quality

MODIS AOD data are used, based on quality assurance (QA) flags, where only retrievals with QA flags of "good" or "very good" are included. In addition, data points with cloud fractions greater than 0.8 or lacking adjacent retrievals are discarded to minimize cloud contamination, following the recommendations of earlier studies (Anderson et al., 2005; Zhang and Reid, 2006; Hyer et al., 2011; Shi et al., 2011) .

MIDAS has been validated against ground-based AERONET data, showing a strong correlation (R = 0.89) and only a slight

positive bias of 2.7% on a global scale (Gkikas et al., 2021). This demonstrates that the MIDAS product accurately represents dust aerosols, especially over key dust regions like North Africa, the Middle East, and Asia. The MIDAS dataset also offers an improvement over existing datasets by providing finer spatial resolution and more accurate dust-specific retrievals, while from its establishment it has supported a lot of studies focusing on dust climatology (Gkikas et al., 2022) and trends (Logothetis et al., 2021), model simulations in respect to emission and transport of dust particles (Kiriakidis et al., 2023) and dust impact

on solar energy production (Masoom et al., 2021; Papachristopoulou et al., 2022). We exploit the MIDAS columnar DOD product at 550 nm, while CALIPSO provides observations at 532 nm. Given that pure dust is characterized by Ångström exponent values close to zero, the difference between 550 nm and 532 nm is expected to be negligible. For this study, we analyze the 550 nm DOD at a 1°×1° spatial resolution over the 10-year period (2007–2017), ensuring consistency with the CALIPSO mission, as MODIS-Aqua lowered its orbit relative to CALIPSO in early 2018, affecting their observational overlap.



### 2.3 POLDER-3 and GRASP/Components

The POLDER-3 (Polarization and Directionality of Earth's Reflectances) sensor, onboard the PARASOL (Polarisation and Anisotropy of Reflectances for Atmospheric Science coupled with Observations from a Lidar) satellite, was developed to improve the characterization of aerosol optical properties through multi-angle, multi-spectral, and polarimetric observations (Steinmetz et al., 2005; Chen et al., 2020; Dubovik et al., 2021; Zhang et al., 2021). POLDER-3 collected radiance measurements across six spectral channels (443, 490, 565, 670, 865 and 1020 nm) and polarization measurements at three channels (490, 670, and 865 nm), allowing it to provide unique insights into aerosol properties by capturing data from up to 16 viewing angles per pixel (Dubovik et al., 2011). This multi-angle capability facilitated global coverage every two days, offering valuable information about the size, shape, and composition of aerosol particles (Dubovik et al., 2011).

The Generalized Retrieval of Aerosol and Surface Properties (GRASP) algorithm, applied to POLDER-3 data, derives detailed aerosol and surface properties, with a component-based approach enhancing the characterization of aerosol properties (Dubovik et al., 2011). In this framework, aerosols are modeled as mixtures of various components—such as hydrated soluble, black carbon, brown carbon, and iron oxides—representative of different aerosol species and characterized by specific chemical compositions and known refractive indices (Dubovik et al., 2006; Schuster et al., 2016). The GRASP/Component algorithm provides daily retrievals of aerosol optical properties like aerosol optical depth (AOD), Ångström exponent (AE), fine-mode AOD (AODF), coarse-mode AOD (AODC), single scattering albedo (SSA) and refractive indices as well as detailed information on aerosol composition and particle size distribution in 0.1°x0.1° or 1°x1° spatial resolution from 2005 to 2013 (Li et al., 2019; Dubovik et al., 2021).

In terms of validation, the GRASP/Component method has demonstrated strong agreement with AERONET stations, particularly for AOD, with negligible bias over both land and ocean (Li et al., 2019; Zhang et al., 2021). Fine-mode and coarse-mode AOD retrievals similarly align well with AERONET, and SSA and AE retrievals from GRASP exhibit high accuracy, enhancing the reliability of aerosol property assessments on a global scale (Li et al., 2019; Dubovik et al., 2021; Zhang et al., 2021). By utilizing advanced radiative transfer calculations without reliance on look-up tables, the GRASP/Component method has achieved high accuracy in aerosol property retrievals (Dubovik et al., 2021). In the present study, we exploit the L3 daily POLDER-3/GRASP-Component AOD retrievals at a 1°×1° spatial resolution for the period when the CALIPSO and PARASOL satellites were flying in tandem in the A-Train constellation (June 2006 – early December 2009). Since POLDER-3/GRASP-Component AOD is not provided exactly at 532 nm, we interpolate between the available wavelengths to derive AOD at 532 nm for consistency with CALIPSO observations.

### 2.4 AERONET

AERONET (AErosol RObotic NETwork) is a global network with over 1,000 automated sun-photometers (Holben et al., 1998; 2001) that measure aerosol columnar optical and microphysical properties in different wavelengths. In this study, we utilize the sun-direct retrievals from AERONET in order to assess the impact of the CALIOP default versus the updated dust



LR values on the columnar AODs (Dubovik et al., 2000; 2006). Specifically, we exploit the latest version (V3) of high-quality, Level 2.0 AERONET data, which includes observations at the highest available temporal resolution (i.e., all data points) (Giles et al. 2019; Sinyuk et al., 2020). To align the spectral columnar AOD (440, 675, 870, 1020 nm) from the AERONET stations

with the AOD at 532 nm from CALIOP-CALIOP, we apply the following Ångström formula (Wagner and Silva, 2007):

$$AOD_\lambda = AOD_{\lambda 0} \cdot \left(\frac{\lambda_0}{\lambda}\right)^{\text{å}\left(\frac{\lambda_0}{\lambda}\right)} \tag{1}$$

, where $\lambda$ refers to the 532 nm, $\lambda_0$ to the 440 nm and å is the Ångström exponent across the 440-675 nm wavelength range.

## 3 Methodology

Our region of interest (ROI) encompasses the major and most active deserts of the planet situated in North Africa and in the

broader region of the Middle East (Ginoux et al., 2012). The selected desert regions are presented in Fig. 1a as defined in Ginoux et al. (2012). ROI hosts major dust sources, primarily over North Africa (yellow-shaded area), including the Bodélé Depression in the northern Chad Basin, the eastern Libyan Desert, the Nubian Desert, and some regions in western Africa. In the Middle East (purple-shaded area), significant sources include the Arabian Desert, with the Rub' al Khali and the Mesopotamian region between the Tigris and Euphrates, as well as deserts in the western parts of Central Asia, notably the

Karakum and Kyzylkum (Ginoux et al., 2012; Hamidi et al., 2013; Gkikas et al., 2022). Subsequently, based on the CALIPSO aerosol subtype product, we focus on dust cases, and finally we adjust the dust LR such as the CALIOP DOD at 532 nm to match the columnar DODs from MIDAS and POLDER-3/GRASP. Dust is the most well-characterized subtype of the CALIPSO aerosol product (Burton et al., 2013), due to the accurate depolarization measurements of CALIOP, which enable the detection of non-spherical dust layers. There are also very good validation metrics for the AOD products of MIDAS and

POLDER-3/GRASP from evaluation studies against AERONET stations (Dubovik et al., 2011; Li et al., 2019; Gkikas et al., 2021; Zhang et al., 2021). The collaboration of these active and passive sensors of the A-Train constellation can lead to the derivation of noteworthy dust LR patterns over the most well-known dust regions around the globe.



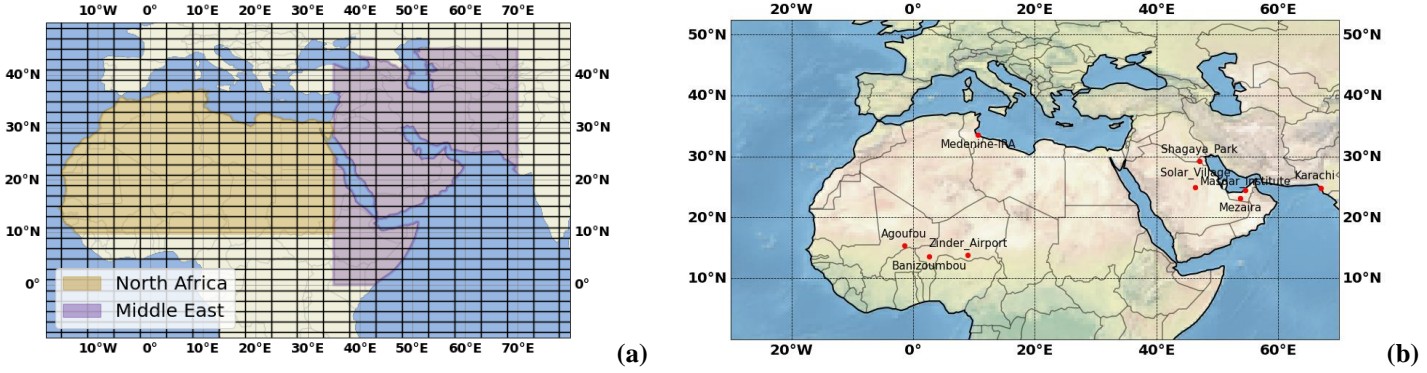

**Figure 1: (a) The North Africa (yellow-shaded area) and Middle East (purple-shaded area) desert regions as defined based on Ginoux**
**et al. (2012). The domain is divided into grids with 2ºx5º spatial resolution grids based on Winker et al. (2013) in order to minimize**
**the signal error in CALIPSO mean profiles, and (b) the AERONET stations over North Africa and Middle East used for evaluating**
**CALIOP DODs.**

### 3.1 Definition of dust cases

Initially, the domain is divided into grids with 2ºx5º spatial resolution, following the recommendations of the CALIPSO team

for reducing the signal error in the mean vertical profiles (Winker et al., 2013). The part of the orbit residing within the desert

regions is identified (Fig. 2a) and then it is separated into its parts falling inside each predefined 2ºx5º grids. For the part of

the orbit falling inside each grid the following criteria need to be satisfied in order for an overpass to be assigned as a dust

case:

(i)      dust records (based only on the aerosol subtype product of CALIPSO) should be at least the 95% of the total

aerosol records,

(ii)     there should be no disturbance in mean profiles due to filtering processes (we want uniform extinction profiles

without any vertical cut-off due to filtering along the overpass),

(iii)    there should be no clouds detected (clear-sky conditions), and

(iv)     the laser beam should be penetrating throughout the atmosphere reaching the ground level.

Overall, 6,915 dust cases fulfill the aforementioned criteria. In Figure 2, the applied methodology is schematically explained

for a random orbit overpassing a specific grid (center of the grid→lat:22.00, lon:15.5). For the part of the CALIPSO orbit

residing within the specific grid (area defined with the purple vertical lines in Fig. 2b), the backscatter coefficient at 532 nm

(Fig. 2c) is spatially averaged in order to be multiplied with the CALIOP LR to finally derive the profile of the extinction

coefficient and the columnar DOD at 532 nm (through the vertical integration of the extinction coefficient).



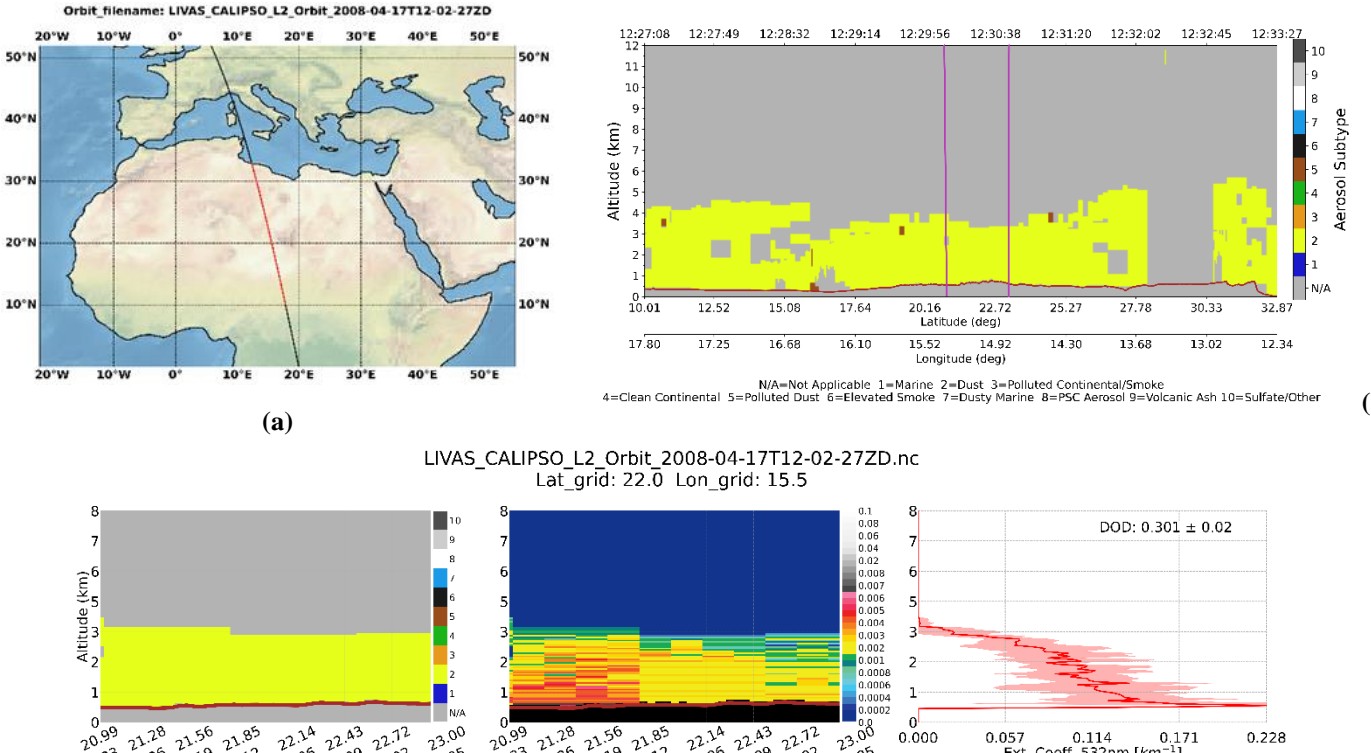

**Figure 2: (a) A single CALIPSO orbit (2008-04-17T12-02-27ZD) over North Africa (the part of the orbit falling inside the desert region is colored red), (b) the respective aerosol subtype curtain plot for the red part of the orbit, and (c) the part of the orbit falling inside a specific grid (center of the grid→lat:22.00, lon:15.5), defined with the two purple vertical lines in (b). The CALIPSO V4 aerosol subtype product (left panel) indicating the presence of pure dust (N/A: not an aerosol layer, 1: marine, 2: dust, 3: polluted continental/smoke, 4: clean continental, 5: polluted dust, 6: elevated smoke, 7: dusty marine, 8: PSC aerosol, 9: volcanic ash, 10: sulfate/other) and the backscatter coefficient 532 nm [$km^{-1}\cdot sr^{-1}$] (central panel). Vertical profile of the spatially averaged extinction coefficient 532 nm (right panel) along with its standard deviation (red-shaded area) for the respective orbit, along with the corresponding columnar DOD based on CALIPSO dust LR (44 sr).**

### 3.2 Match-up methodology between A-Train constellation

After the identification of the dust cases based on CALIPSO, we identify the coincident 1ºx1º products from MIDAS (DOD) and POLDER-3/GRASP (AOD) datasets. For MIDAS, we check the 10-year period spanning from 2007 to 2017, while for the POLDER-3/GRASP, the collocated period starts in June 2006 and ends in December 2009, as this marks the point when the PARASOL satellite's orbit was lowered to under the A-Train constellation. Additional to MIDAS and CALIPSO DOD, as long as we focus on the solely presence of dust cases based on CALIOP aerosol subtype algorithm, and we search for coincident POLDER-3/GRASP products, we can refer to DOD instead of the term of AOD for its retrievals.

The match-up procedure is thoroughly presented in Fig. 3, while the filters that are applied on the 1ºx1º MIDAS and POLDER-3/GRASP grids for quality assurance control and for the extraction of coastal pixels are outlined in Table 1. A random 2ºx5º





predefined grid is presented (purple square), divided into 10 subgrids with 1ºx1º spatial resolution (red squares), which is the resolution of MIDAS and POLDER-3/GRASP products utilized in the current study. For a specific orbit (purple points), the grey shaded 1ºx1º grids are the ones coincident with the CALIPSO overpass. For these grids to be included in the mean

columnar DOD computation from MIDAS and POLDER-3/GRASP products, they must meet the criteria outlined in Table 1. The "LandPercentage" and "Land_sea_Flag" filters are applied to POLDER-3/GRASP and MIDAS products respectively, to exclude coastal pixels. Additionally, a threshold of 0.05 for "ResidualRelative" (the mean root square of relative error from fitting POLDER-3 non-polarised radiance measurements by the GRASP algorithm) is applied as recommended in several studies (Dubovik et al., 2011; Chen et al., 2020; Zhang et al., 2021) to ensure more reliable aerosol retrievals. For MIDAS

product, no additional criteria need to be employed in the filtering process, as long as during its development only high-quality data were used (Gkikas et al., 2021). If there is at least a single 1ºx1º grid fulfilling the aforementioned criteria we process to the computation of the mean columnar DOD from MIDAS and POLDER-3/GRASP products and these values are assigned to the 2ºx5º predefined grid.

**Table 1: Filters applied on POLDER-3/GRASP and MIDAS products to select quality-assured data while ensuring the elimination**
**of coastal regions.**

| POLDER-3/GRASP | "ResidualRelative" < 0.05 |
| --- | --- |
| | "LandPercentage" = 100 |
| MIDAS | "Land_sea_Flag" = 1 |

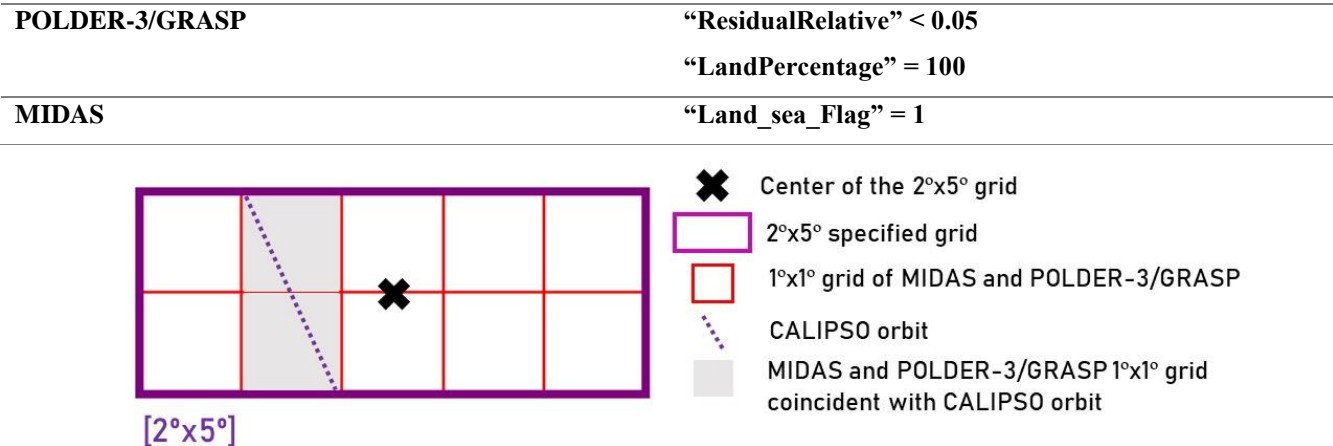

**Figure 3: Schematic representation of match-up methodology between 2ºx5º predefined CALIPSO grids and the 1ºx1º resolution of MIDAS and POLDER-3/GRASP products.**

**3.3 Derivation of dust lidar ratio**

The match-up methodology described in the previous paragraph led to 5,591 and 1,817 collocated pairs between CALIOP-MIDAS and CALIOP-POLDER-3/GRASP, respectively. As a first step for the derivation of LR, the DOD differences are computed based on the equation:

$$\text{DOD}_{\text{diff,i}} = \text{DOD}_{\text{i}} - \text{DOD}_{\text{CALIPSO}} \quad (2)$$

, where $i$ refers to MIDAS or to POLDER-3/GRASP, respectively. For the adjustment of LR, the North Africa and Middle

East $\text{DOD}_{\text{diff,i}}$ distributions are filtered separately using the Interquartile Range (IQR) method. This method can be applied to





various distributions, from symmetric ones (e.g., normal distribution) to skewed ones, as it focuses on the central 50% of the data and is less influenced by extreme values. The following equation is used for its calculation (Wackerly et al., 2008):

$$IQR = Q3 - Q1 \ (3)$$

, where the Q1 is the first quartile ($25^{th}$) and the Q3 is the third quartile ($75^{th}$). In order to define the outliers, the following

equations are used for the lower and upper threshold:

$$lower\ threshold: Q1 - 1.5 \cdot IQR \ (4)$$

$$upper\ threshold: Q3 + 1.5 \cdot IQR \ (5)$$

The IQR method, commonly applied in satellite data processing for robust filtering (Zhang et al., 2019; Bhattarai et al., 2024), is implemented on the $DOD_{diff}$ restricting the adjusted LR values in "reasonable" ranges ($53.1 \pm 7.9$ sr for Saharan dust and

$37.4 \pm 5.3$ for Middle Eastern dust - values summarized in Floutsi et al., 2023), as long as too high or low values of $AOD_{diff}$ could lead to unrealistic LR retrievals. The IQR method yielded a final collection of 5,203 and 1,773 cases over both North Africa and Middle East for the synergy of CALIOP-MIDAS and CALIOP-POLDER-3/GRASP, respectively.

After the filtering of the $DOD_{diff}$, in order to estimate the dust LR per 2ºx5º grid we want for each of the collocated pairs to adjust the LR in order for the $DOD_{CALIOP}$ to be equal either to $DOD_{MIDAS}$ or to $DOD_{POLDER-3/GRASP}$, following the equation:

$$LR_{updated,i} = LR_{CALIPSO} \cdot \left(DOD_{diff,i} - 1\right) \ (6)$$

, where $i$ refers either to MIDAS to POLDER-3/GRASP products. The $LR_{CALIPSO}$ represents the default LR value used for dust layers within the CALIPSO algorithm and it is equal to 44 sr (Kim et al., 2018). Higher values than 44 sr are assigned to the dust $LR_{updated}$ if $DOD_{CALIOP}<DOD_{MIDAS\ or\ POLDER-3/GRASP}$ and lower ones in the opposite case. Finally, for each 2ºx5º grid a mean value for dust LR is computed, along with its standard deviation, while the counts (number of dust cases) used for each grid

towards the adjustment of LR are also calculated.

**3.4 Evaluation against AERONET**

To evaluate the impact of dust LR on CALIOP's columnar DOD retrievals, we first aim to identify dust cases near AERONET stations during the CALIPSO mission, which spans the period from 2006 to 2023. Following Moustaka et al. (2024), we begin by identifying CALIPSO orbits that pass within a 100 km radius of AERONET stations located in the desert domains outlined

in Fig. 1a. These stations must have available measurements within a ±30 min window centered on the satellite's overpass time. Along each CALIPSO overpass, we ensure that all the criteria mentioned in Section 3.1 are met. Based on Amiridis et al. (2015), we also require: (i) the difference between the AERONET station elevation and CALIPSO's mean track elevation to be less than 100 m, and (ii) a relatively flat ground (slope less than 400 m) along the satellite overpass. By applying these additional criteria, we minimize DOD biases caused by the different conditions that could be sampled by the ground and

satellite-based instruments. For the part of the CALIPSO orbits falling inside the 100 km radius around the AERONET station, the respective dust LR values of the 2ºx5º grids around the ground site are applied on the backscatter coefficient retrievals at





532 nm in order to derive the columnar DOD at 532 nm. Under the previous constraints, a total number of 135 dust cases were identified in 9 AERONET stations, most of them over Middle East and the region of Sahel, as presented in Fig. 1b.

## 4 Results

### 4.1 Examination of DOD retrievals among the A-Train constellation

Firstly, we emphasize on the DOD comparison between CALIPSO versus POLDER-3/GRASP and MIDAS after filtering the outliers in the $DOD_{diff}$ distributions and prior the computation of the "updated" dust LRs. In Figure 4, we present the box-whisker plots of the $DOD_{diff}$ (y axis) between the two synergies of CALIOP-POLDER-3/GRASP (Fig. 4a) and CALIOP-MIDAS (Fig. 4b) divided into 10 equal-sized bins by CALIOP DODs (x axis). The obtained statistics of the linear regression are provided separately for North Africa (yellow color) and Middle East (purple color), including the slope of the linear regression, the Mean Bias Error (MBE), the correlation coefficient (R) and the Root Mean Square Error (RMSE), along with the number of collocated samples per region (counts). Based on the statistics, there is a pronounced underestimation and overestimation of CALIOP DODs over North Africa (MBE equal to 0.086 and 0.075 for CALIOP-POLDER-3/GRASP and CALIOP-MIDAS synergy, respectively) and Middle East (MBE equal to -0.021 and -0.022 for CALIOP-POLDER-3/GRASP and CALIOP-MIDAS synergy, respectively), respectively, indicating that higher dust LRs over North Africa and lower over Middle East could improve the DOD CALIOP retrievals. This outcome is in agreement with previous studies, relying on ground-based lidar observations, reporting that dust LRs between these two major desert regions are different due to the dust particles mineralogy (Müller et al., 2007; Groβ et al., 2011; Preiβler et al., 2011; Tesche et al., 2009a; Tesche et al., 2011a; Kanitz et al., 2013a; Baars et al., 2016; Rittmeister et al., 2017; Kaduk, 2017; Haarig et al. 2017a; Urbanneck ,2018; Bohlmann et al., 2018; Filioglou et al., 2020; Szczepanik et al., 2021; Haarig et al., 2022). In both synergies, the underestimation over the Middle East becomes more pronounced for higher CALIOP DODs (over Arabian Peninsula), while there is also the need for DOD increment for some smaller- to moderate-DODs, possibly due to the inclusion of some central Asian deserts (eastern of Caspian Sea) into the domain (Fig. 1a), as noted also by Kim et al. (2020) and discussed in detail in Section 4.2. In contrast, over North Africa, most of the discrepancies on CALIOP DODs are recorded for the smaller- to moderate- dust loads, while for the CALIOP-MIDAS synergy, a slight overestimation is observed for the high DODs, which can be attributed to some sub-Sahel regions (Section 4.2).





(a)

(b)

**Figure 4: Box-whisker plots for the Dust Optical Depth differences (DOD_diff) (y-axis) (displaying the 75/25 percentiles at the boxes, and the median in the center line, while the whiskers are displaying the ±1.5 IQR from the 25/75 percentiles, respectively) between (a) CALIOP – POLDER-3/GRASP and (b) CALIOP-MIDAS, along the CALIOP DODs (x-axis). The yellow boxes are assigned for North Africa, while the purple ones for Middle East. The background color denotes the over-/underestimation (red/blue color). The number of collocated measurements, the correlation coefficient (r), the slope of the linear regression, the Mean Bias Error (MBE), and the Root Mean Square Error (RMSE) scores are provided separately for North Africa and the Middle East.**



## 4.2 Geographical distribution of dust lidar ratio

The DOD$_{diff}$ in Fig. 4 were used for the adjustment of dust LR following Eq. (6). For each 2º×5º grid over North Africa and Middle East domains (Fig. 1a) a mean dust LR value was computed, along with its standard deviation, the number of dust cases used for the LR adjustment. The LR geographical distribution of the differences between the upgraded dust LR (obtained via the CALIOP-POLDER-3/GRASP and CALIOP-MIDAS synergies, respectively) and the default value (44 sr) assigned to dust aerosols in the CALIPSO retrieval algorithm were also estimated. For the 10-year period of the CALIOP-MIDAS synergy, both annual and seasonal LR maps were produced. In contrast, for the CALIOP-POLDER-3/GRASP synergy, only annual LR maps could be generated due to the significantly shorter period of coincident retrievals between the PARASOL and CALIPSO satellites (1,773 instead of 5,203).

In Fig. 5, the mean dust LR map from the CALIOP-POLDER-3/GRASP synergy is illustrated. Dust LRs are maximized (up to ~53 sr) over or near the strongest dust source globally (Washington et al., 2009; Ginoux et al., 2012), namely the Bodélé Depression in the northern Chad basin. Other North African desert regions yielding high dust LRs include the Grand Elg of Bilma and the basin of the Aïr (both over Niger), the Libyan desert, and regions in Sudan. On the western side of North Africa, notable areas are the Erg El Djouf (over Mali and Mauritania) and the Grand Erg Occidental (over Algeria), all characterized by LR values mostly in the range of 47–52 sr. Over the Arabian Peninsula, our results agree well with the dust LR range (37.4 ± 5.3 sr) obtained from ground-based lidar measurements (Müller et al., 2007; Kaduk, 2017; Urbanneck, 2018; Filioglou et al., 2020; Floutsi et al., 2023). Even in the absence of dust LR values from ground-based measurements in the literature reported for the central Sahara, and over the major dust sources (e.g. Bodélé), the values reported in the literature (53.1 ± 7.9 sr) from campaigns (SAMUM 1,2 – Ansmann et al. (2011); SALTRACE – Weinzierl et al. (2017)) studying the optical and microphysical properties of dust particles originating from North Africa after short- and long-range transport are within the range of our CALIOP-POLDER-3/GRASP dust LRs. The regions over the Karakum and Kyzylkum deserts (central Asia) can be considered as a "blind" spot for dust LR, while the only study in the literature from Hofer et al. (2020) reported values in the range of 35-40 sr over Dushande (Tajikistan) in the framework of the CADEX (Central Asian Dust EXperiment) campaign (Althausen et al., 2019). Additionally, according to the Fig. S1 and S2, most of the dust LR values are characterized by low standard deviation values (mainly less than 5 sr) and a sufficient number of cases, while over the Arabian Peninsula, the standard deviations are higher comparable to the ones over the North Africa (principally in the range of 5-7 sr) even for a significant number of dust cases. This is possibly attributed to either the transport of dust burdens from the Saharan and Central Asian regions (Rezazadeh et al., 2013; Al-Hemoud et al., 2022; Gkikas et al., 2022), leading to a mixing of dust particles with higher and lower LR values (different mineralogy - Caquineau et al., 2002; Scheuvens et al., 2013; Ahmady-Birgani et al., 2019) or to misclassification issues in CALIPSO aerosol subtype product with some aerosol layers erroneously classified as dust. In the same manner, regions over the Sahel, could include some misclassified smoke layers from fires over Central



African, with the plumes generally moving towards northern African regions (i.e. the Sahel, sub-Sahel and the Guinea Gulf) (Groβ et al., 2011).

In Fig. 6 the mean dust LR map from the synergy CALIOP-MIDAS is presented. Similar to the CALIOP-POLDER-3/GRASP synergy, higher values are observed over North Africa, while the incorporation of MIDAS yields higher values of LR in the
desert regions over the central part of Asia (around Karakum Desert of Turkmenistan and the Kyzylkum Desert of Uzbekistan and Kazakhstan-in the range of 47-55 sr). Except this evident difference between the two synergies, the LR pattern remains almost the same over the North Africa and in the Arabian Peninsula. The Bodélé Depression and the Grand Erg of Bilma once again are characterized by the highest dust LR values (up to 53.4 sr), followed by the Libyan Desert and certain regions in the western part of the Sahara, with values ranging from 48 to 52 sr. Even if the number of cases for the adjustment of dust LR
are significantly increased (Fig. S5) for the CALIOP-MIDAS synergy, due to the 10-year-period of collocated measurements, the standard deviations (Fig. S4) are of the same magnitude with the CALIOP and POLDER-3 synergy, with lower values over North Africa and central parts of Asia and higher values over the Arabian Peninsula.

For the synergy of CALIOP-MIDAS the seasonal dust LR maps, their standard deviations, along with the number of cases per 2º×5º predefined grid are also presented in Fig. 7 and Fig. S7-8. During JJA, the LR distribution is characterized by the most
fragmented pattern, with the highest dust LR values scattered across the desert belt extending from the western part of North Africa and the Arabian Peninsula to Central Asia. Under the prevalence of strong surface winds and low-level jet streams, dust production and transport from the most well-known dust sources is intensified (Al-Hemoud et al., 2022; Gkikas et al., 2022). Conversely, during DJF, dust production diminishes, with the dust LR hotspots restricted mainly over the major dust sources. During MAM and SON, the LR distribution has a transitional pattern between the more active (JJA) and the less active seasons
(DJF) for dust emission and transport. According to the seasonal standard deviation of dust LR (Fig. S7), the higher values are confined to the Middle East. These elevated values can be partly attributed to misclassification issues in the CALIPSO aerosol classification scheme. This is because the region is significantly contaminated with aerosol types originating primarily from anthropogenic sources over the Arabian Peninsula. The CALIOP algorithm has difficulty distinguishing pure dust from mixtures of dust with smoke or pollution, particularly when the mixture's higher depolarizing nature is predominantly driven
by dust particles. However, the significantly lower variability of dust LR over the Middle East during the DJF underscores the importance of dust transport from adjacent sources, particularly from Sudan, during the summer months. This transport is influenced by prevailing wind patterns, such as northeasterlies and northwesterlies, which carry a substantial load of dust layers into the Middle East (Rezazadeh et al., 2013).



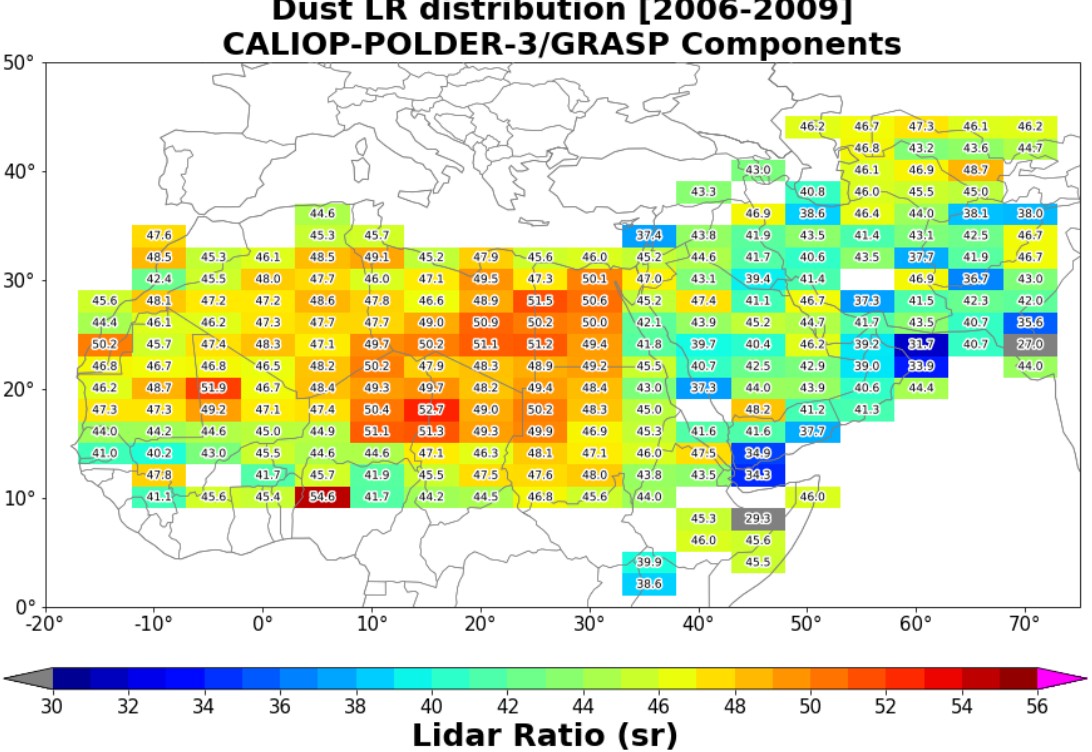

**Figure 5: The spatial variability of dust Lidar Ratio (LR) based on the synergy of CALIOP and POLDER-3/GRASP. The values represent the mean LR per 2°x5° predefined grid.**





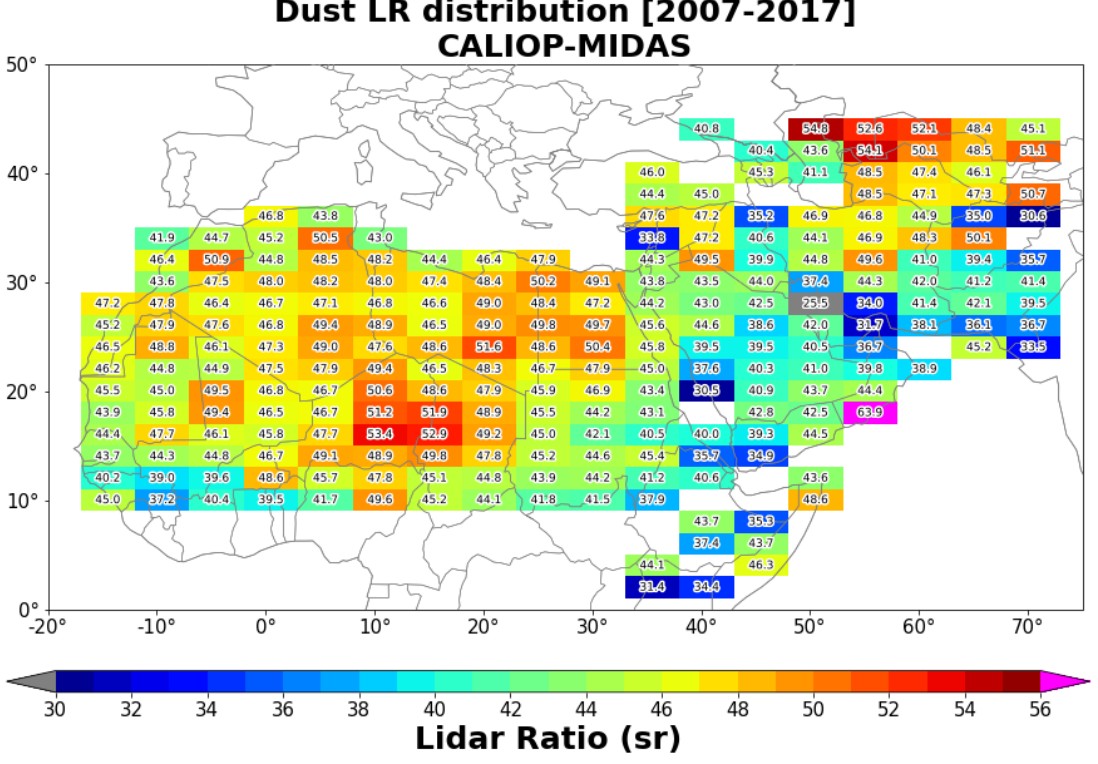

**Figure 6: The spatial variability of dust Lidar Ratio (LR) based on the synergy of CALIOP and MIDAS. The values represent the mean LR per 2°x5° predefined grid.**





**Figure 7: The seasonal distribution of dust Lidar Ratio (LR) for (a) DJF, (b) MAM, (c) JJA and (d) SON based on the synergy of CALIOP and MIDAS. The values represent the mean LR per 2°x5° predefined grid.**

## 4.3 Evaluation of the LR effect on CALIOP's DOD retrievals against AERONET

In this section we are assessing CALIOP's performance in terms of reproducing DODs against AERONET ground-based retrievals. Specifically, we are examining the influence of the newly established dust LRs on the agreement between CALIPSO and AERONET, in contrast to the default value of 44 sr. The CALIPSO-AERONET collocation methodology discussed in detail in Section 3.4, while Fig. 1b shows the AERONET stations in which the defined criteria have been fulfilled. The obtained DOD evaluation metrics, including the Mean Bias Error (MBE) and the Root Mean Square Error (RMSE), are depicted in Fig. 8a and 8b, respectively. Each colored bar corresponds to the different LRs, namely the default one (44 sr, red bar) and those derived from the CALIOP-POLDER (green bar) and CALIPSO-MIDAS (blue bar for the annual and brown for the seasonal) synergies. Furthermore, the dashed color lines featuring star symbols correspond to the LRs differences (upgraded – default; right y-axis) as these have been computed from each combination of the satellite instruments. The different shaded areas refer to the stations over North Africa (yellow-shaded area) and the Middle East (purple-shaded area), as defined in Fig. 1a. For





each AERONET site (bottom x-axis), along with the evaluation scores, we also present the number of matchups between
spaceborne and ground-based retrievals (top x-axis). Note that for the seasonal CALIPSO-MIDAS synergy the stations
Banizoumbou, Medenine-IRA and Karachi are excluded since the seasonal collocation against AERONET didn't yield any
significant number for cases, due to missing values of LR over a number of 2°x5° predefined grids. Specifically, for the seasonal
collocation against AERONET, based on the seasonal LR maps (Fig. 7) different LR values are applied in respect to the season
(e.g. when a specific CALIPSO orbit passes over an AERONET station in June, the LR from Fig. 7c is applied to the satellite's
retrieval for the derivation of the columnar DOD). Our findings indicate that across all stations and for the CALIOP-POLDER
and CALIOP-MIDAS synergies during their overlapping periods, it is evident that the upgraded LRs lead to lower MBEs,
with respect to those found when the default dust LRs is considered, thus indicating a better agreement between CALIPSO
and AERONET (Fig. 8a). This "improvement" coincides with a reduction of the RMSEs in 7 out of 9 AERONET sites. In
most of the selected sites, the CALIPSO-AERONET agreement improves when the MIDAS DOD is considered for the
calculation of the upgraded LR (blue bars) instead of the POLDER-3/GRASP AOD (green bars). However, it is noteworthy
that both CALIPSO-MIDAS and POLDER-3-CALIPSO combinations indicate similar LR tendencies almost in all sites, which
are necessary for minimizing CALIPSO-AERONET departures under dust scenes. It is reminded that negative LR differences
indicate that the reduction of the default value (mainly required at sites in the Middle East) can improve the agreement between
CALIPSO and AERONET. A reverse condition is valid for the AERONET sites situated in North Africa (i.e., an increase in
dust LR is required). As expected, the modifications of both MBE and RMSE are highly dependent on the magnitude of the
dust LR departures. This is attributed to the fact that the default/upgraded dust LRs are applied on the dust backscatter retrievals
for the derivation of the dust extinction coefficient and finally of the dust optical depth (via the integration of the vertical dust
extinction coefficient profile). Both MBEs and RMSEs exhibit a further decrease in Agoufou, Mezaira and Solar Village when
the dust LRs are computed on a seasonal basis via CALIPSO-MIDAS synergy (indicated by the brown bar). Overall, the
proposed adjustments of the dust LR lead to better evaluation scores. However, persistent discrepancies between CALIPSO
and AERONET remain, as previously documented in the literature (Amiridis et al., 2013; Kim et al., 2018; Moustaka et al.,
2024). These deficits are primarily attributed to the inability of the CALIOP sensor in detecting tenuous layers, particularly
during daytime due to the strong solar background illumination, as well as to misclassification issues in the CALIPSO aerosol
classification scheme (Winker et al., 2009; Burton et al., 2012; Ma et al., 2013; Li et al., 2021).



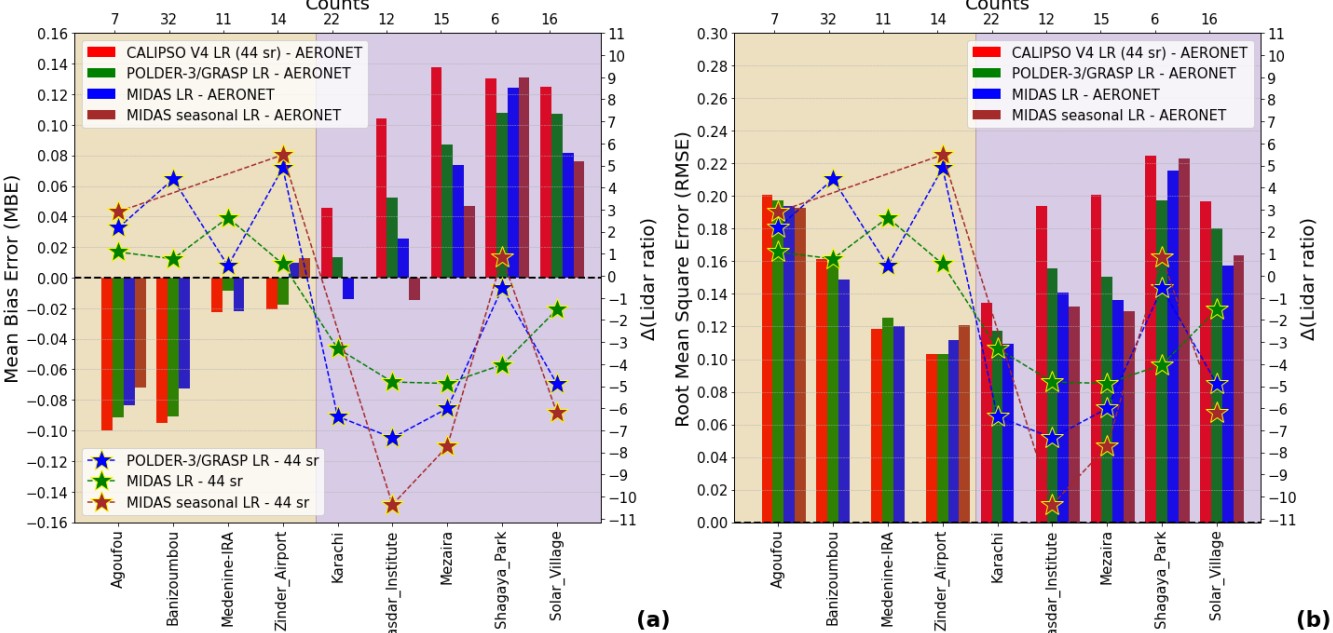

**Figure 8: Bar plots of (a) Mean Bias Error (MBE) and (b) Root Mean Square Error (RMSE), from comparisons between AERONET and CALIOP DOD by applying the default CALIPSO dust LR of 44 sr (red bars), the updated LR from the synergy between CALIOP-POLDER-3/GRASP synergy (green bars), and the updated LR from the annual and seasonal synergy between CALIOP-MIDAS synergy (blue bars and brown bars, respectively) on CALIOP's backscatter retrievals. On the bottom x-axis the AERONET stations used in the evaluation are presented, while in the top x-axis are the counts of the dust cases used for the derivation of the metrics per station. The second y-axis represents the LR$_{diff}$ between the CALIOP default dust LR (44 sr) versus the dust LR from the synergies of CALIOP-POLDER-3/GRASP (green stars) and CALIOP-MIDAS (blue stars for annual; brown stars for seasonal), respectively.**

## 4.4 DOD climatology

In this section, we evaluate the impact of the revised dust LRs on dust climatology across a range of spatiotemporal scales. For the DOD analysis covering the period from 2007 to 2022, we apply on CALIOP's dust backscatter retrievals, gridded at 1° x 1° spatial resolution, the default dust LR as well as those resulting from the synergies of CALIOP-POLDER-3/GRASP (annual basis) and CALIOP-MIDAS (both on an annual and seasonal basis). The vertical profiles of the dust backscatter coefficient have been derived via a well-established discrimination technique (Tesche et al., 2009b), which effectively separates dust from non-dust components in dusty mixtures (polluted dust and dusty marine - Kim et al. 2018). The validity of the discrimination technique has been justified in numerous past works (Amiridis et al., 2013; Freudenthaler et al., 2009; Nisantzi et al., 2014; Mamouri and Ansmann, 2014; 2017; Hu et al. 2020; Wang et al., 2022; Xiao et al., 2023; Panahifar et al., 2023). In our study, we also apply certain modifications to the depolarization ratio thresholds, as detailed in the recent work by Moustaka et al. (2024). These thresholds are derived from the DeLiAn database (Floutsi et al., 2023) and are set at 0.28 for dust and 0.03 for





polluted continental/smoke aerosols. For marine layers, the thresholds vary depending on humidity conditions (Haarig et al., 2017b), with values of 0.01 for wet and 0.15 for dry environments. It is important to be clarified that all CALIPSO orbits intersecting the two ROIs are processed without employing the criteria outlined in Section 3.1. Moreover, we are excluding from our analysis all grids in which the upgraded dust LRs have been computed based solely on a single collocated dust case

(see supplementary Figs. S2, S5 and S8) between the different sensors (CALIOP, POLDER-3, MODIS).

### 4.4.1 Annual and Seasonal DOD geographical patterns

In Fig. 9, we present the spatial distribution of CALIPSO DOD at 532 nm, representative over a 15-year period (2007-2022), computed based on its default algorithm (a-i), in which a constant value of 44 sr is applied on the backscatter retrievals for the derivation of the columnar DOD. Concurrently, we have reproduced the temporal averaged maps utilizing the revised dust

LRs (a-ii to a-iv), derived from the CALIOP-POLDER-3/GRASP and CALIOP-MIDAS synergies. The DOD differences between the DODs computed with the updated LRs versus the CALIPSO default DODs are also presented on the right column of Fig. 9. Furthermore, in Fig. 10 we display the corresponding seasonal DOD patterns during (a) DJF, (b) MAM, (c) JJA and, (d) SON. The first two columns correspond to the DOD retrievals with (i) the CALIPSO default dust LR and (ii) with the seasonal LRs obtained from the CALIOP-MIDAS synergy, while in the last column we present the DOD departures, driven

by the LR adjustments for each synergistic approach, as presented in Figs. S3, S6 and S9.

Based on the default CALIPSO retrieval algorithm, the most intense dust loads (DODs up to 0.5) are recorded in the western Sahara, in Saudi Arabia, along the coastal regions of Iran as well as in the southern part of Pakistan. In other areas, DODs do not exceed 0.3, except in Iraq, where DODs can reach up to 0.4 (Fig. 9 a-i). Through the implementation of the upgraded dust LRs, the spatial patterns of DOD are modified remarkably. In North Africa, a zone of high DODs (up to 0.6) stretching from

Mauritania to Chad and it is bounded between 10° N and 25° N is visible in all possible synergies between the active (CALIPSO) and passive (POLDER-3/MODIS) spaceborne instruments. Within this zone, two "spots" of maximum DODs in Mali-Mauritania and in Niger-Chad are recorded. An interesting finding is the almost two-fold increase of DOD in the Bodélé Depression, the most active dust source of the planet (Washington et al., 2009), when the adjusted dust LRs are considered. Due to the increase of dust LR, the updated CALIOP DODs are less underestimated than those (0.8-1.2) given in Gkikas et al.

(2022). Similarly, the upgraded DODs (0.6-0.7) in the El-Djouf Desert (Mauritania-Mali) match better with those found in Gkikas et al. (2022). Across Algeria, Libya and Egypt, as well as in Soudan, slightly higher DODs are derived through the CALIOP-POLDER-3/GRASP synergy (Fig. 9 a-ii) compared to those calculated by combining CALIOP and MIDAS on an annual basis (Fig. 9 a-iii). Within the zone of high DODs, the two peaks are more distinguishable when the new LRs have been calculated from the collocated CALIOP-MIDAS sample on a seasonal basis. Moreover, areas yielding moderate DODs (0.3-

0.4) are recorded in Libya and in Egypt. The increase of DODs in North Africa, after the implementation of the upgraded dust LRs, is evident throughout the year, as illustrated in Fig. 10. The most evident differences between the default and the upgraded DOD spatial distributions are encountered in Niger-Chad becoming maximum in JJA. During summer, the activation of dust sources (Ginoux et al., 2012; Hu et al., 2019) in the western Sahara Desert is depicted both when the default (Fig. 10 c-i) and



the adjusted (Fig. 10 c-ii) LRs are considered. Notably, during the boreal summer (Fig. 10 c-ii), a large part of the western

Sahara experiences significant dust loadings (DOD > 0.5) in agreement with other studies in the literature (Pu and Ginoux, 2018; Song et al., 2021; Gkikas et al., 2022; Tindan et al., 2023). In addition, the updated DODs are higher and align closely with those (0.7–0.8) reported by Gkikas et al. (2022) and Song et al. (2021), respectively.

Over the Arabian Peninsula, the spatial extension of moderate DODs (0.3-0.4), observed in the default CALIPSO patterns (Fig. 9 a-i), is shrank on the corresponding maps generated with the adjusted dust LRs (Figs. 9 a-ii, 9 a-iii, 9 a-iv). Among the

three synergies, similar DOD fields are drawn between the CALIOP-POLDER-3/GRASP (Fig. 9 a-ii) and CALIOP-MIDAS (Figure 9 a-iv; seasonal) combinations. In both patterns, the highest DODs (up to 0.5) are recorded in the Rub' al Khali Desert (southern sector of S. Arabia) in agreement with Gkikas et al. (2022) and Tindan et al. (2023). In addition, the lowest DODs across the Arabian Peninsula are obtained from the CALIOP-MIDAS synergy (Fig. 9 b-ii). As can be seen in Figs. 9 a-ii to iv, the annual DOD patterns over the Arabian Peninsula are highly variable in spatial terms (i.e., Mesopotamia) due to the

pronounced LR diversity over the region, as discussed in Section 4.2. On a seasonal basis (Fig. 10), the spatial variability of the dust burden in the Arabian Peninsula is commonly reproduced, when the default (first column) and the revised (second column) dust LRs are applied, in contrast to what has been found over North Africa. However, it is evident that there are discrepancies between the seasonal maps produced with the default and the adjusted dust LRs. These discrepancies are recorded on the DOD magnitude, which is higher based on the default CALIPSO lidar ratio with respect to those derived from

the revised values (i.e. CALIPSO-MIDAS synergy). Similar to North Africa, DODs also peak over the Arabian Peninsula during the JJA, with values predominantly above 0.4, consistent with findings from other studies in the literature (Pu and Ginoux, 2018; Song et al., 2021).

Based on the default CALIPSO dust LR, dust loads over the Karakum (Turkmenistan), the Kyzylkum (Uzbekistan) Deserts, in the Ustyurt Plateau (Li and Sokolik, 2018) and in the Garabogazköl Basin (Shen et al., 2016) yield low optical depths (<

0.15) (Fig. 9 a-i). Their intensity increases after the implementation of the revised LRs with positive departures ranging from 0.05 to 0.1 (CALIOP-POLDER-3/GRASP; Fig. 9 b-i) and from 0.05 to 0.25 (CALIOP-MIDAS; Figs. 9 b-ii and 9 b-iii). These DOD increments are mainly observed during summer when the dust activity maximizes over the area (Elguindi et al., 2016; Shen et al., 2016; Li and Sokolik, 2018; Song et al., 2021; Gkikas et al., 2022) in agreement with other studies in the literature (Pu and Ginoux, 2018; Gkikas et al., 2022; Tindan et al., 2023). Declining DOD tendencies are recorded along the eastern

coasts and the southern sector (Dasht-e Margo Desert) of Iran, in Afghanistan and in the southern parts of Pakistan.

The regional maps of the DOD departures (i.e., the right column in Fig. 9 and the third column in Fig. 10) can effectively demonstrate the effect of the adjusted dust lidar ratios on the CALIPSO dust optical depth levels. The analysis reveals a discernible pattern in the DOD departures, underscoring the necessity for adjustments to the CALIPSO dust lidar ratio to enhance the congruence between CALIOP and either POLDER-3/GRASP or MIDAS. Across North Africa, positive departures

predominate attaining maximum magnitude primarily in proximity to dust sources. A transition zone is evident between the Sahara and the Arabian Peninsula, characterized by almost neutral DOD biases, as evidenced by the CALIOP-MIDAS synergy. In most parts of the Arabian Peninsula, the predominance of relatively low negative DOD biases indicates that the default dust



LR (44 sr) should be slightly lower. In contrast to the CALIOP-MIDAS combinations, the signal of the DOD biases derived via the CALIOP-POLDER synergy it is not uniform.


**Figure 9: Geographical distribution of the annual Dust Optical Depth (DOD) at 532 nm based on (a-i) the CALIPSO default LR, (a-ii) the LR from the synergy of CALIOP-POLDER-3/GRASP and (a-iii, a-iv) the annual and seasonal LR from the CALIOP-MIDAS synergy. The DOD differences between the CALIPSO default product and the DODs derived from the synergies of (b-i) CALIOP–POLDER-3 and (b-ii, b-iii) CALIOP–MIDAS are also presented in the right column.**




**Figure 10: Geographical distribution of the seasonal Dust Optical Depth (DOD) at 532 nm during (a) DJF, (b) MAM, (c) JJA and (d) SON, based on (i) the CALIPSO default LR and (ii) the seasonal LR from the synergy of CALIOP-MIDAS, while in the 3rd column (iii) the DOD differences between the two DOD retrievals are also presented separately for each of the 4 seasons.**




### 4.4.2 DOD inter-annual and intra-annual variability

An assessment of the adjusted dust LRs on the CALIOP DODs at a regional scale was also performed. Figure 11 illustrates
the inter-annual (i) and the intra-annual (ii) DOD timeseries, representative for the 2007-2022 period, separately for North
Africa (a; yellow shaded areas in Fig. 1a) and the Middle East (b; purple shaded areas in Fig. 1a). The regional DODs have
been reproduced based on the standard dust LRs (red line) as well as with the corresponding values derived by the CALIOP-
POLDER synergy (green line), the CALIOP-MIDAS synergy (blue line) and the CALIOP-MIDAS synergy applied on a
seasonal basis (brown line). As expected, there is positive shift for DODs (ranging from 0.05 to 0.15) over North Africa (Fig.
11 a-i) when the adjusted dust LRs are implemented, due to the positive DOD departures dominating across the Sahara Desert
(Figs. 9 and 10). The seasonal DOD cycle is almost commonly reproduced among all approaches with minima and maxima
DODs during winter and in June, respectively (Fig. 11 a-ii). However, there are differences in the regional DODs. Specifically,
the lowest levels are obtained when the default dust LR (red line) is used, with DODs ranging from 0.1 to 0.34. The
implementation of the dust LRs provided by the CALIOP-MIDAS synergy on an annual basis (blue line) results in an increase
of the DODs from 0.05 to 0.08. Slightly higher DODs are found based on the CALIOP-POLDER-3 synergy (green line). These
minor DOD deviations found between the MIDAS-based and the POLDER-based LRs can be interpreted by the fact that for
MIDAS the DOD is used instead of the AOD utilized in the CALIOP-POLDER-3 synergy. The derivation of dust LR based
on the seasonally collocated CALIOP-MIDAS sample (brown line) leads to higher DODs, compared to the default CALIPSO
values (Fig. 11 a-ii), similarly to what has been shown for the other two combinations. Nevertheless, the increments vary with
time and are maximum during the dry period of the year, reaching up to 0.13, while they are suppressed during winter (up to
0.05).

In the Middle East (bottom panel in Fig. 11), the consideration of the adjusted dust LRs leads in a slight reduction of the default
regional CALIOP DODs, throughout the study period (Fig. 11 b-i). In terms of magnitude, the regional DODs based on the
default CALIPSO aerosol scheme varies from 0.05 to 0.45 whereas the maximum values are significantly reduced in 2019,
2020 and 2021. The systematic negative DOD offset is better evident in the intra-annual variation plot (Fig. 11 b-ii), which
also shows that the three curves (red, green and blue) covariate within the course of the year. On an annual basis, DODs are
minimized in December whereas the maximum levels are recorded in June (Fig. 11 b-ii). The intra-annual DOD pattern slightly
changes when the seasonal dust lidar ratios derived from the CALIOP-MIDAS synergy are applied (brown curve in Fig. 11 b-
ii). More specifically, the DOD peak shifts from June to May and decreases from 0.35 to 0.33, whereas the minimum DODs
are observed in November instead of December. Furthermore, the updated DODs are slightly higher/lower than the default
levels in May/March and identical in April. The DOD peaks recorded in either May or June have also been reported by Gkikas
et al. (2022) and in Song et al. (2021). Nevertheless, the differences found between the findings of the current work and those
reported in the aforementioned studies, apart from the different study periods, are due to the different regions considered. In
this study, the regional DODs have been calculated considering the entire Middle East region instead of its specific parts, as
has been presented in Gkikas et al. (2022) and Song et al. (2021).







**Figure 11: Interannual (i) and interseasonal (ii) variability of Dust Optical Depth (DOD) at 532 nm, representative for the period 2007–2022, regionally averaged over (a) North Africa and (b) Middle East. The red color represents the DOD computed with the CALIPSO default LR, while green, blue and brown colors are used for the implementation of the updated LR values based on CALIOP-POLDER-3/GRASP and CALIOP-MIDAS annual and seasonal synergies, respectively.**

## 5 Summary and conclusions

The CALIPSO satellite mission has provided unprecedented state-of-the-art aerosol and cloud observations over a 17-year period (2006-2023), significantly improving our knowledge of various aerosol-related processes relevant to a wide range of environmental applications. The CALIOP lidar, mounted on the CALIPSO satellite, has delivered aerosol and cloud vertical profiles, up to the lower stratosphere, probing areas characterized by diverse atmospheric constituents. One of the key advantages of the CALIOP lidar is the deployment of a depolarization channel enabling the detection of non-spherical particles (e.g., dust, volcanic ash, ice crystals). Previous studies have documented that the CALIPSO aerosol retrieval algorithm outperforms when mineral dust particles are probed in the atmosphere. In elastic lidars (such as CALIOP), for the derivation



of the aerosol extinction coefficient from the backscatter coefficient, it is required a factor known as lidar ratio. For dust

aerosols, the lidar ratio is set to 44 sr in the CALIPSO aerosol retrieval algorithm and is constant worldwide. However, ground-based lidar measurements acquired at different places of the world have shown that the dust LR varies in space and time. Driven by this fact, in the present study we focus on the estimation of the dust LRs over North Africa and Middle East, encompassing the most active dust sources of the planet. Our analysis relies on a synergistic approach by combining CALIPSO vertical dust profiles (a set of strict criteria has been implemented to ensure the presence of only mineral dust) with quality-

assured columnar passive aerosol retrievals derived from the MIDAS dataset (MODIS-based) and the POLDER-3/GRASP satellite observations. All spaceborne instruments are/were part of the A-Train constellation thus enabling coincident observations. The study period covers the entire CALIPSO era, spanning from June 2006 to August 2023. We have calculated the updated (revised) dust LRs within the study domain by collocating: (i) CALIOP-POLDER-3/GRASP during the period 2006-2009, (ii) CALIOP-MIDAS during the period 2007-2017 and (iii) CALIOP-MIDAS, on a seasonal basis, over the period

640    2007-2017.

The dust LR maps produced display a significant spatial variability over both North Africa and in the Middle East. According to our findings, the highest dust LRs are recorded over and near the Bodélé Depression in the northern Chad basin (values up to 53.4 sr), over the Libyan desert (48-52 sr) and over dust-affected areas eastwards of the Caspian Sea (47-55 sr). Over the Arabian Peninsula, the adjusted dust LRs range from 40 to 44 sr and are mostly lower than the default (standard) value (44 sr).

This outcome is consistent with the findings reported in other studies. The major difference between the CALIOP-POLDER and CALIOP-MIDAS synergies is that the latter highlights higher LR values over central Asia, particularly over the Karakum and Kyzylkum deserts, which are less pronounced in the CALIOP-POLDER results. Based on the CALIOP-MIDAS synergy on a seasonal basis, it is found that the highest dust LRs are obtained in summer, while the lowest ones are recorded in winter, indicating that the proposed adjustments follow the dust optical depth seasonal cycle. In order to evaluate our methodology,

we have identified 135 cases where pure dust loads are probed by CALIOP in the surrounding area (<100 km) of 9 AERONET stations. The updated dust LR yielded better evaluation metrics against AERONET stations in comparison with the CALIPSO default LR, in terms of the mean bias error (MBE) and the root mean square error (RMSE). In particular, the best improvements were recorded over Karachi, Masdar Institute and Mezaira, with each synergy (combination) performing better or worse over the different stations, while the deviations from the DOD based on the CALIPSO default LR were more significant over

stations where the LR showed a notable decrease or increase from the default value of 44 sr.

At the final stage of the analysis, it has been assessed the effect of the upgraded dust LRs on the climatological DODs. By implementing the calculated dust LRs, the spatiotemporal DOD patterns are significantly modified compared to those produced by considering the default LR. According to the standard CALIPSO aerosol scheme, DODs are maximized (up to 0.5) over the western Sahara and in Saudi Arabia, while similar levels are recorded along the coasts of Iran and in the southern Pakistan.

Based on the revised dust LRs, in North Africa, a distinct zone of elevated DODs reaching values of up to 0.6, extends from Mauritania to Chad, spanning the latitudinal range of 10° N to 25° N. This zone is consistently observed across all synergies between active (CALIPSO) and passive (POLDER-3/MODIS) spaceborne instruments. Within this region, two prominent





hotspots of maximum DODs are identified, one located in the Mali-Mauritania area and the other in the Niger-Chad region. In Saudi Arabia, the implementation of the adjusted dust LRs results in more heterogeneous DOD fields in comparison to the default CALIPSO retrieval algorithm. In summary, our findings suggest that the revised dust LRs lead to an increase in the intensity of dust loads across North Africa (maximized over/nearby the major dust sources) while declining DOD tendencies are found in the Arabian Peninsula. Furthermore, a narrow transition zone of neutral DOD declinations (using the default CALIPSO DODs as reference) between North Africa and the Arabian Peninsula is obtained from the CALIOP-MIDAS synergy. Finally, the revised dust LRs increase the DOD over regions east of the Caspian Sea, contrary to what has been found in Iran, Afghanistan and Pakistan.

This study focuses on a dynamic (in terms of space and time) estimation of the dust lidar ratio over the most active deserts globally, via a synergistic approach involving active (CALIPSO-CALIOP) and passive (MIDAS and POLDER-3) spaceborne retrievals. Our research aims to enhance the efficacy of CALIPSO dust retrievals, thereby increasing their utility in a broad range of scientific studies. The analyses performed can be extended to other large deserts in the northern hemisphere (e.g. Taklamakan), as well as to the geographically limited and less active dust sources in the southern hemisphere. In a follow-up study, we will investigate the dust-induced direct radiative effects based on radiative transfer model (RTM) simulations. This will involve the use of the updated CALIPSO dust profiles, the intensive spectral optical properties (including single scattering albedo, asymmetry parameter, and refractive index) obtained by the GRASP/POLDER-3 retrieval algorithm for the identified dust cases, along with the spectral surface albedos sourced from the HAMSTER dataset (Roccetti et al., 2024). Given the strong dependence of dust LRs on the composition of the mineral particles (Ansmann and Müller, 2005), the incorporation of the mineralogical maps derived by the EMIT instrument (Connelly et al., 2021; Coleman et al., 2024) will strengthen the interpretation of the outcomes obtained in the current study. One of the challenging tasks encountered in our analysis was the assessment of the proposed adjustments to the default CALIPSO dust lidar ratio. Due to the scarcity of ground-based measurements within the study region, the evaluation was limited to the vicinity of a few AERONET stations. A better justification and refinement of our calculations will be achieved via the incorporation of the vertically resolved lidar ratio observations acquired by the ATLID lidar, onboard the EarthCARE satellite (Illingworth et al., 2015; Wehr et al., 2023). Finally, among the major improvements to be implemented in the upcoming CALIPSO retrieval algorithm (Version 5), one is related to the use of new aerosol models for the estimation of the lidar ratio, underscoring the necessity for a more accurate representation of this critical parameter.

*Data availability.* The LIVAS dust products are available upon request from Vassilis Amiridis (vamoir@noa.gr), Emmanouil Proestakis (proestakis@noa.gr), and/or Eleni Marinou (elmarinou@noa.gr). The MIDAS dataset has been developed in the framework of the DUST-GLASS project (grant no. 749461; European Union's Horizon 2020 research and innovation program under the Marie Skłodowska-Curie Actions) and is available at https://doi.org/10.5281/zenodo.4244106 (Gkikas et al., 2020). The POLDER-3/GRASP Components products are publicly available at the GRASP-OPEN website (https://www.grasp-



open.com/products/polder-data-release/). The Aerosol Robotic Network (AERONET) data are publicly available and can be accessed through the official website at: https://aeronet.gsfc.nasa.gov/.

*Author contributions.* Conceptualization, A.M. and A.G.; methodology, A.M., A.G., S.K., V.A.; software, A.M.; validation, A.M., E.P. and A.G.; writing—original draft preparation, A.M; writing—review and editing, A.M., A.G., E.P., A.L., S.K.,
O.D., K.T., C.Z., V.A.; funding acquisition, A.G., S.K. All authors have read and agreed to the published version of the manuscript.

*Financial support.* This work was supported both from COST Action CA21119 Harmonia: International network for harmonisation of atmospheric aerosol retrievals from ground based photometers, supported by COST (European Cooperation in Science and Technology) and the Hellenic Foundation for Research and Innovation (H.F.R.I.) under the "2nd Call for
H.F.R.I. Research Projects to support Post-Doctoral Researchers" (Project Number: 544)

*Acknowledgment.* We are grateful to AERONET for providing high-quality sun-photometer observations, calibrations, and products. We would like also to acknowledge the use of POLDER data "POLDER/PARASOL Level-1 data originally provided by CNES (http://www.icare.univ-lille1.fr/) and AERIS/ICARE Data and Services Center, processed by Cloudflight Austria GmbH with GRASP software (https://www.grasp-open.com) developed by Dubovik et al. (2011, 2014, 2021) and Li et al.
710 (2019).

*Competing interests.* V. Amiridis is in the editorial board of AMT and coordinating the special issue that the paper is submitted.

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
