# Peer review of "Enhancing dust aerosols monitoring capabilities across North Africa and the Middle East using the A-Train satellite constellation"

_EGUsphere, 2025_

## Author Comment (AC1)

**Replies to Referee #1**

We thank the Referee for his/her thorough review and encouraging assessment of our manuscript. We have carefully addressed each of the points raised, and we believe that the suggested revisions have helped to substantially improve the clarity and overall quality of the manuscript. Below, we provide a detailed, point-by-point response (in red) to the Referee's comments (in black).
* * *
**General comments:**

This manuscript presents an important contribution to the field of aerosol remote sensing by enhancing the estimation of dust lidar ratios (LRs) through a synergistic use of multiple satellite datasets (CALIOP, MODIS/MIDAS, and POLDER-3/GRASP). The research primarily addresses a critical limitation in the CALIPSO dust retrieval algorithm—the use of a fixed dust lidar ratio (LR) of 44 sr.

The effort to improve the accuracy of CALIOP-derived Dust Optical Depth (DOD) in North Africa and the Middle East is commendable and of clear relevance to both the scientific community and stakeholders.

A noteworthy contribution of the study is the clear demonstration that regional variability in dust properties necessitates the use of spatially and seasonally resolved LR values. The analysis reveals that higher LR values are required over North Africa (~53 sr), while lower values (~37 sr) are more appropriate for the Middle East. These adjustments lead to measurable improvements in DOD retrievals, particularly in regions like the Bodélé Depression and the western Sahara, which are known for intense dust activity.

The authors further validate their findings against AERONET sun-photometer measurements, reporting substantial reductions in both Mean Bias Error (MBE) and Root Mean Square Error (RMSE). Seasonal analysis also reveals strong correspondence between revised LR values and observed dust activity patterns, highlighting the method's robustness across varying climatic conditions.

We sincerely thank the Referee for the positive and encouraging feedback. We are pleased that the proposed methodology and its contributions to improving CALIOP-derived DODs were well received. We appreciate the recognition of the regional and seasonal variability in dust lidar ratios and the significance of addressing this limitation in the CALIPSO retrieval algorithm. We also thank the Referee for acknowledging the validation efforts and the robustness of the method across different regions and seasons. This valuable feedback is highly motivating and reinforces the relevance of our work for the aerosol remote sensing community.

*While the methodology is generally robust and the findings are well-supported, I would like to offer several constructive observations that may help refine the study and guide future work as follows.*

1. Although the improved dust LRs clearly lead to enhanced agreement with satellite and ground-based AOD measurements, the broader implications for climate modeling and radiative forcing are not fully explored. A quantitative analysis of the impact on surface energy budgets, direct radiative effects, or global dust forcing would elevate the study's significance, therefore I suggest to write a statement for future work related to this.

Thank you for this insightful comment. We agree that a quantitative assessment of the impact of revised dust LRs on surface energy budgets and radiative forcing would significantly enhance the broader relevance of our work. While such an analysis is beyond the scope of the current study, we have included a statement in the manuscript to highlight this as a direction for future research. Specifically, in the **Summary and conclusions** section (lines 667–671), we mention:
*"In a follow-up study, we will investigate the dust-induced direct radiative effects based on radiative transfer model (RTM) simulations. This will involve the use of the updated CALIPSO dust profiles, the*

*intensive spectral optical properties (including single scattering albedo, asymmetry parameter, and refractive index) obtained by the GRASP/POLDER-3 retrieval algorithm for the identified dust cases, along with the spectral surface albedos sourced from the HAMSTER dataset (Roccetti et al., 2024).''*
We hope this addresses the reviewer's concern and demonstrates our awareness of the broader implications of our findings.

2. The authors reference the relevance of their results for upcoming missions (e.g., EarthCARE) and future algorithm updates (CALIPSO v5). While promising, the manuscript does not provide a detailed roadmap for how the improved LRs will be operationally integrated. Outlining a pathway or collaboration with mission teams could enhance the practical value of the study.

Thank you for your constructive comment. We assume that the Referee is referring to the part of the manuscript where we discuss the relevance of our results considering upcoming satellite missions and algorithm updates (see Section 5, lines 676–681). Specifically, we would like to clarify that our intention is not to suggest an operational integration of the derived dust lidar ratios from our study into the EarthCARE or CALIPSO v5 algorithms. Rather, our aim is to highlight the value of our findings as a benchmark for future intercomparisons.
For instance, the EarthCARE mission will provide vertically resolved lidar ratio profiles through the ATLID instrument, and these observations will offer a valuable opportunity to compare against our regionally and seasonally adjusted lidar ratios. Likewise, the upcoming CALIPSO Version 5 algorithm will include updated aerosol models with new lidar ratio values. In this context, our work underscores the importance of using regionally refined lidar ratios and can contribute to the broader discussion on algorithm improvement, rather than being directly integrated into these systems.
We have revised the manuscript accordingly to clarify this point and avoid potential misunderstandings. The modified part now states:

"*A better assessment and validation of our findings will be possible through comparison with the vertically resolved lidar ratio observations to be provided by the ATLID lidar onboard the EarthCARE satellite (Illingworth et al., 2015; Wehr et al., 2023). Finally, one of the key updates in the forthcoming CALIPSO Version 5 retrieval algorithm involves the adoption of new aerosol models for lidar ratio estimation, further highlighting the need for an improved and regionally representative characterization of this critical parameter.*"

In conclusion this work represents a valuable contribution to aerosol remote sensing, offering practical improvements to satellite dust retrievals in one of the most important source regions globally. Addressing the aforementioned limitations in future research would not only strengthen the current findings but also broaden the applicability of the proposed methodology across other regions and satellite missions.

**Specific comments:**

Page 7 in paragraph 220-please provide a reference for the interpolation procedure to derive AOD at 532nm.

Thank you very much for your comment. The original statement on interpolation was brief, but we have now provided a more detailed explanation of the equations applied for the interpolation within the POLDER-3/GRASP section. The common part of the AERONET section has been transferred to the POLDER-3/GRASP section and modified accordingly. We hope these changes make everything clearer.

Based on the applied modifications, the differences between the interpolation procedures for the two sections (POLDER-3/GRASP Components and AERONET) are as follows:

1. In the case of POLDER-3/GRASP Components, the Ångström exponent is estimated from the provided AODs (Equation 1), and then the AOD at 532 nm is estimated (Equation 2).

2. In the case of AERONET, the Ångström exponent is already provided, and only Equation 2 is applied.

3. For POLDER-3/GRASP, the Ångström exponent is estimated for the 440–670 nm wavelength range, whereas for AERONET, the Ångström exponent is provided for the 440–675 nm wavelength range.

Figure 3-page 11- the shades of red/purple are very close and not easily distinguishable-please choose contrasting colors.

We sincerely apologize for the color choices in Figure 3. We have updated the colors to make them more easily distinguishable, and we have also increased the font size of the legend. You can now compare the previous version of Figure 3 with the updated one. We hope this clarifies the issue.

**Figure 3 before modification:**

[Figure]

**Figure 3 after modification:**

[Figure]

Figure 4 page 14 – in caption for the correlation factor, better use "R".

Thank you very much for noticing. In the figure, we use the letter "R" to represent the correlation coefficient, but in the legend, we mistakenly used "r"

Page 15 line 398 -I think could be useful to write in here when you refer to Figure S1 and S2 to specify that they could be found in the supplement material.

We apologize for overlooking the reference to the Supplement. We have modified the specified part as follows:
"Additionally, as shown in Figs. S1 and S2 (provided in the Supplement), most of..."

Figure 8 page 21- Very difficult to follow all information in panels a and b;  too much information; For me it could be much better if you put on separate panels the graphs related to annual/seasonal.The shades of red and brown for me are very confusing- difficult to distinguish; I would change brown with a different color-maybe black or grey.

Thank you for your valuable comment, and we apologize for any inconvenience caused. In response, we have modified the red and brown colors to orange and cyan, respectively, to improve visual clarity and resolve the difficulty in distinguishing between the curves. All figures in the manuscript have been adjusted using the **Copernicus color blindness simulator** to ensure accessibility for readers with color vision deficiencies.

While we appreciate your suggestion to separate the annual and seasonal analyses into different panels, we have chosen to present all relevant information in a single plot to facilitate direct comparison. Displaying the data side by side allows the reader to better evaluate the impact of using upgraded seasonal LR values (cyan bars) in contrast with both the annual LR values (green and blue bars) and the default CALIPSO V4 LR value of 44 sr (orange bars).

**Figure 8 before modification:**

[Figure]

**Figure 8 after modification:**

[Figure]

Line 500- when you refer to depolarisation thresholds are you referring that the value is set for 0.28 for dust? In general when reading the word "thresholds" I would look for an interval between max and min values. Please clarify.

Thank you for your comment. We have replaced the term "threshold" with the term "value" and revised the section of the manuscript describing the separation of the dust mixtures to improve clarity and comprehension.

Line 510 Add respectively after "derived from the CALIOP-POLDER-3/GRASP and CALIOP-MIDAS synergies".

Thank you for the addition. We have modified the sentence as follows: "Concurrently, we have reproduced the temporally averaged maps utilizing the revised dust LRs (a-ii to a-iv), derived from the CALIOP-POLDER-3/GRASP and CALIOP-MIDAS synergies, respectively."

Line 514- I have noticed you have used the word "departure" several times when you refer to the difference between two values- not sure if it is correctly used- please check with an English speaker.

Thank you for your comment. We have replaced the word "departure" throughout the manuscript and made the necessary adjustments. For instance, in some cases, we opted for the word "differences" which is more commonly used in remote sensing literature.

Lines 528-529- "Within the zone of high DODs, the two peaks are more distinguishable when the new LRs have been calculated from the collocated CALIOP-MIDAS sample on a seasonal basis." I do not agree with this statement. That area looks very similar in the 2 cases. At least is not \'more distinguishable\' for me.

We apologize for the confusion. You are correct. Based on Figure 9a-iv, the peaks **are not more distinguishable** when the seasonal LR maps are applied to CALIOP's backscatter retrievals for estimating the annual DOD. This sentence was likely carried over from a previous version and was used to describe the increment of DOD hotspots in Figure 10 over North Africa when the CALIOP-MIDAS seasonal LR maps are implemented. We removed the sentence.

Figure 11- Again, I have difficulties to distinguish red and brown curves.

Thank you for your comment. In response to your feedback, we have changed the red and brown curves to orange and cyan, respectively, to improve their visibility and make them more easily distinguishable. To ensure accessibility, we verified the updated colors using the Copernicus color blindness simulator.

**Figure 11 before modification:**

[Figure]

**Figure 11 after modification:**

[Figure]

---

## Author Comment (AC2)

**Replies to Referee #2**

We sincerely thank the Referee for his/her thorough and insightful evaluation of our manuscript. We greatly appreciate the constructive comments and the positive feedback on the scientific value and clarity of our work. The points raised—particularly concerning Equation 6, the methodology presentation, and the data visualization—have been extremely helpful in identifying areas for improvement and clarification. We have carefully addressed each of the comments below and made the necessary revisions to strengthen the manuscript. We are confident that the changes have significantly improved the quality and readability of the paper. Below, we provide a detailed, point-by-point response (in red) to the Referee's comments (in black).
* * *
**General comments:**

The paper by Moustaka et al. describes the synergistic use of long-term CALIOP vertical aerosol profiles and two passive satellites to adapt the CALIOP lidar ratio choice for aeolian mineral dust. They replace the standard lidar ratio selection algorithm by a variable synergistic approach which scientifically improves the aerosol extinction retrievals obtained from CALIOP in the vicinity of dust and thus provide a data set which can be used to estimate radiative forcing more accurately.

The authors adapt the lidar ratio in a way that the total AOD (as for dust dominated scenes only, equivalent to the Dust Optical Depth) fits to the one retrieved by the satellite-based imagers (POLDER and MODIS). By doing so, they retrieve a map of dust lidar ratio across two research areas (North Africa and Middle East). The results are validated against selected AERONET stations in the region of interest. Furthermore, they apply these new results (lidar ratios) to CALIOP backscatter profiles to obtain enhanced dust optical depth climatologies in the research regions and compare them to the ones obtained with the standard CALIPSO retrieval.

The paper is highly relevant and of great interest for the research community and very well written. Usually, I would have only minor comments, but as I am struggling with Equation 6, which is essential for the whole publication, I have to propose "major revisions" (see major comment below). I suspect that the "Equation 6 issue" can be solved fast or is just a misunderstanding. In this case, the paper can be published after addressing the other comments. If there is a substantial mistake in the retrieval (what I doubt given the presented results), the results need to be newly calculated and interpreted.

Given that the "equation 6 issue" is minor and/or just a typo, the paper is suitable as a highlight paper, as it gives a very important improvement in the global data set of dust load in the atmosphere. However, for that reason, it would be great if the authors could make the newly derived products available for the research community, i.e., the CALIOP results with adapted lidar ratio retrieval.

We sincerely thank the Referee for the constructive review and for the overall very positive assessment of our manuscript. We are grateful for the recognition of the novelty and value of our synergistic approach to adapt the CALIOP lidar ratio for dust, as well as for the acknowledgment of the methodological improvements and their potential impact on climate studies through enhanced dust optical depth (DOD) retrievals. The Referee's feedback, especially regarding the scientific relevance, clarity of the manuscript, and the robustness of the results, is deeply appreciated. We are particularly encouraged by the suggestion that the work is suitable as a highlight paper, which motivates us further to refine and clarify our methodology. The issue raised regarding Equation 6 indeed stems from a typographical mistake, which we clarify and correct in the revised manuscript. We have also addressed all other comments-major and minor-with detailed revisions and responses, as outlined below.

**Major/General comments:**

1. Equation 6 cannot be correct: According to Eq. 2, the absolute difference between the DODs is calculated and then used in Eq. 6. However, this would lead to negative adapted lidar ratios in case the DOD differences are less than 1.

If one assumes a let's say DOD difference of 0.1, Equation 6 would lead to: 44sr * (0.1 -1) = -39.6 sr as updated lidar ratio which cannot be possible because it is negative.

A DOD difference of 2 would lead to an updated lidar ratio of 44 , i.e. the same as already used. And no difference would lead to a negative updated LR of -44.

Please check and comment. I am confident that it can be clarified fast, because this formula actually cannot lead to the results presented later on.

Thank you very much for your careful reading and helpful comment. You are absolutely right — the equation as written was incorrect, and this was due to a typo from a previous manuscript version when we had mistakenly set the $DOD_{diff}$ equal to $DOD_{CALIPSO} - DOD_i$.

In our analysis we wanted to linearly scale the lidar ratio in order to drive the DOD difference to zero, following the logic:

$$\Delta LR = LR_{CALIPSO} \cdot DOD_{diff} \text{ (I)}$$
$$LR_{updated} = LR_{CALIPSO} + \Delta LR \text{ (II)}$$

Combining (I) and (II) yields:

$$LR_{updated} = LR_{CALIPSO} \cdot (1 + DOD_{diff}) \text{ (III),}$$

where $LR_{CALIPSO}$ is equal to 44 sr (Kim et al., 2018). For example, a DOD difference of 0.15 would result in:

$$LR_{updated} = 44 + 6.6 = 50.6 \text{ sr,}$$

which is consistent with the goal of compensating for the underestimation.

We appreciate your attention to this detail — it helped us correct the formulation and clarify the intent. The Equation 6 within the manuscript will be updated accordingly. Please note that the results presented were based on the correct version of the equation.

2. With respect to the correction of this issue, the methodology section should also be updated with some formulas for non-lidar experts on how to retrieve the DOD (AOD) from CALIOP. I.e., lines 265 to 270 are the most important part of the methodology but were yet just briefly described (especially when comparing to all the other extensive descriptions). For non-lidar experts this might not be obvious. Thus, these very important 5 lines should be more highlighted and even extended, i.e. by a sketch and showing the formulas (I.e., the (D)AOD is the integrated (dust)bsc times the Lidar ratio written as formula could help and you later can always refer to it.)

Thank you again for your valuable feedback. In response to your suggestion, we've updated the methodology section, particularly lines 265 to 270, to provide a more detailed and accessible explanation for non-lidar experts. We have expanded on the retrieval of DOD from CALIOP, emphasizing the key formulas and providing additional context to ensure clarity.

Specifically modified the part of the manuscript as follows:

"Overall, on the basis of CALIOP observations in the period June 2006-August 2023 over the regions of interest, 6,915 dust cases were identified fulfilling the aforementioned criteria. In Figure 2, the

applied methodology is schematically explained for an indicative CALIPSO granule and for a specific grid area over the Sahara Desert (center of the grid→lat:22.00, lon:15.5). Initially, for the part of the CALIPSO L2 5 km aerosol profiles residing within the grid area (confined between the magenta vertical dashed lines in Fig. 2b), the backscatter coefficient at 532 nm profiles (Fig. 2c) are spatially averaged in order to derive the mean backscatter coefficient vertical profile. It should be emphasized that the mean backscatter coefficient profile in the identified cases is computed only on the basis of dust atmospheric layers, as classified by the CALIOP aerosol subtype algorithm (Kim et al., 2018) and does not include the contribution of dust components from dust mixtures (i.e., polluted dust and dusty marine). However, it should be noted that even the atmospheric layers classified as "dust" may still include a relatively small non-dust fraction in the aerosol mixture. Accordingly, the mean dust extinction coefficient at 532 nm profiles are obtained on the basis of the computed mean dust backscatter coefficient at 532 nm profiles through the implementation of predefined aerosol-subtype dependent lidar ratio (LR) values (Eq.3). The LR is defined as the extinction-to-backscatter ratio and is crucial for converting the CALIOP backscatter profiles into extinction profiles. Based on CALIPSO algorithm the LR of dust particles at 532 nm is universally constant and equal to 44 sr (Kim et al., 2018).

$$\sigma(z) = \text{LR}_{\text{CALIPSO}} \cdot \beta(z) \tag{3}$$

where $\sigma(z)$ is the dust mean extinction coefficient at 532 nm profile, $\beta(z)$ is the dust mean backscatter coefficient at 532 nm profile at altitude z, and $\text{LR}_{\text{CALIPSO}}$ is the predefined dust LR of 44 sr at 532 nm. Once the average dust extinction coefficient at 532 nm profile is obtained, vertical integration with respect to height provides the corresponding dust optical depth (DOD) at 532 nm of each identified dust case (Eq.4).

$$\text{DOD} = \int_{z_{\text{bottom}}}^{z_{\text{top}}} \sigma(z)\, dz \tag{4}$$

where $z_{\text{bottom}}$ and $z_{\text{top}}$ represent the lower and upper boundaries of the identified dust layer(s), respectively. It should be noted that the integration is effectively carried out only over the vertical layers of the atmosphere identified as containing dust — potentially spanning multiple non-contiguous layers — while dust-free (clear air) regions in between, above or below the dust layers do not contribute to the DOD.
"

As a reference, the previous version of lines 265–270 was:

"Overall, for each identified case, the backscatter coefficient profiles at 532 nm were averaged spatially within each grid cell, and the corresponding mean extinction coefficient profiles were calculated using a fixed lidar ratio. The integration of the extinction profile over height yielded the Dust Optical Depth (DOD) for each case."
We hope the revised version provides the clarity you suggested, and we thank you again for helping improve the accessibility and completeness of the methodology section.

It should be also emphasized again that you assume that the backscatter profile of CALIPSO to be only representative for dust aerosol as I assume you do no separation of the backscatter in a dust and non-dust component (Later in Section 4.4. you explain it, but this is too late then).

Thank you for your comment. We appreciate your feedback, and we agree it is important to emphasize this assumption earlier in the manuscript.

To clarify, we assume the CALIPSO backscatter profile to be representative of dust aerosol because we select only layers classified as "dust" by the CALIOP aerosol subtype algorithm (Kim et al., 2018). We do not further separate the backscatter into dust and non-dust components within these layers, and dust mixtures (e.g., polluted dust, dusty marine) are not included in our analysis.

Additionally, we had produced a supporting analysis (for internal use) showing that for both MIDAS and POLDER-3/GRASP synergies, the particulate depolarization ratios reported by CALIOP consistently exceed 0.27 (see the following figure), which indicates a strong dust signal with negligible contributions from other aerosol types. This further justifies our assumption that the CALIPSO backscatter profiles under these conditions are representative of dust aerosol.

We have added the following clarification earlier in the manuscript to address this point:

"It should be emphasized that the mean backscatter coefficient profile in the identified cases is computed only from layers classified as 'dust' by the CALIOP aerosol subtype algorithm (Kim et al., 2018) and does not include contributions from dust mixtures (i.e., polluted dust, dusty marine). This approach, supported by consistently high depolarization ratios (>0.27) (not shown here) observed in the selected cases, ensures that the backscatter profiles are representative of dust aerosols."

We hope this clarifies our approach and addresses your concern.

[Figure]

3. Furthermore, the authors "just" adapt the lidar ratio for the retrieval of the extinction coefficient from the given backscatter profile multiplied by the lidar ratio. However, for elastic lidars, the assumption of the lidar ratio is already used when calculating the backscatter coefficient (see, e.g., Klett-Fernald 1984). The authors should briefly discuss what effect an alternated lidar ratio (in the range they have retrieved it) would have on the retrieved backscatter profile to give evidence that there is no need to recalculate the backscatter profiles and the methodology they used is valid.

Thank you for this important comment. To address it, we performed a sensitivity analysis to estimate the error introduced in the backscatter coefficient profiles when using a fixed lidar ratio in the inversion (e.g., 44 sr), while the actual lidar ratio may differ within a realistic range. We simulated the Klett backscatter retrieval error using synthetic extinction and backscatter coefficient profiles. Because the Klett retrieval is sensitive to the vertical aerosol structure, we generated 1000 synthetic extinction profiles with varying random vertical structures, composed of up to three top-hat layers, and a vertical resolution of 100 m to reflect realistic atmospheric variability.

[Figure]

[Figure]

The synthetic extinction profiles were scaled to yield aerosol optical depths (AOD) of 0.1, 0.5, or 1. For this study, we assumed the presence of dust only, so the AOD corresponds to the DOD. For each extinction profile, we simulated the particle backscatter coefficient using 21 constant lidar ratio classes ranging from 34 to 54 sr. These profiles were used as input to a forward lidar simulation to create synthetic attenuated backscatter signals, with a laser tilt of 3° from zenith and a wavelength of 532 nm. We applied the Fernald-Klett-Sasano inversion method to these synthetic signals using a fixed lidar ratio of 44 sr — consistent with the standard CALIPSO V4 processing — to retrieve the backscatter coefficient profiles. We then compared these retrieved backscatter profiles with the "true" synthetic backscatter profiles (generated with the actual lidar ratios) and calculated both the maximum absolute and relative retrieval biases. The molecular backscatter and extinction coefficients used in the forward simulations were calculated using the US Standard Atmosphere (1976) and the ARC software (https://github.com/nikolaos-siomos/arc).

**Maximum absolute bias**

[Figure]

**Maximum relative bias**

This analysis highlighted that the bias introduced in the backscatter retrievals by using a constant lidar ratio of 44 sr — even when the true lidar ratio varies between 34 and 54 sr — remains relatively small and does not significantly impact the shape or magnitude of the resulting backscatter profiles in most realistic scenarios. Therefore, recalculating the backscatter coefficient profiles for each lidar ratio is not

necessary, and our methodology — which multiplies the fixed-backscatter profile with the varying lidar ratios to obtain extinction — remains valid and robust for the purpose of this study.

4. For the result parts (Sec 4.4.), the authors always refer to both approaches using MODIS AND POLDER: However, it would be convenient for the reader if the final statements could be made based on only one of these data sources especially if the results are similar (MIDAS seems to be the choice for that as longer available).

Thank you for your comment. We understand that it might be more convenient for the reader to base the final conclusions on a single dataset. However, as the results from both MODIS and POLDER show similar trends, we believe that it is important to include both, especially since POLDER is essential for our ongoing and future work. Specifically, in our follow-up study, we will use the POLDER/GRASP retrievals to extract spectral optical properties, which are critical for our radiative transfer simulations (RTMs). These simulations will provide insights into the dust-induced direct radiative effects, for which POLDER/GRASP will provide critical inputs. Additionally, it is encouraging that the results from both the MODIS and POLDER datasets are in close agreement. Such consistency across different sensors is a positive indicator that the methods and assumptions used are robust. It would be concerning if different sensors yielded entirely different values for the same property, as it could suggest issues with the retrievals or inconsistencies in data processing.

5. More a hint: The paper is very well written and of high quality, but partly a bit longish and repetitive. Partly there is an overlap and repetition between the data set and methodology section. This does not need to be changed here but should be considered for the next publication to be more concise.

Thank you for your kind feedback. We appreciate your suggestion regarding the length and repetitiveness of the manuscript. We will certainly take this into consideration for future publications, ensuring to streamline the content and reduce any overlaps between sections, particularly between the dataset and methodology. We will aim to make our writing more concise while maintaining clarity and completeness.

6. All data should be made available for the scientific community, also the LIVAS data set AND your novel Calipso dust extinction data set produced in this work. I.e., upload to zenodo or some dedicated database.

Thank you for your suggestion. While we fully support the idea of making data publicly available, the LIVAS dataset is not developed or owned by us, and therefore, we are unable to upload or share it publicly. The data is available upon request from the data providers, as detailed in the Data Availability section: Vassilis Amiridis (vamoir@noa.gr), Emmanouil Proestakis (proestakis@noa.gr), and Eleni Marinou (elmarinou@noa.gr).

As for the novel CALIPSO dust extinction dataset produced in this work, we will create and provide a NetCDF file containing the respective lidar ratio (LR) values across North Africa and the Middle East from both synergies (CALIOP-POLDER-3/GRASP & CALIOP-MIDAS), which will be made available in a public repository.

We hope this approach meets the requirements and addresses the concern.

7. References are partly missing and many typos in the reference section – please check carefully.

We have carefully checked the reference section and have addressed any missing references and typos. Everything should be in order now. Thank you for pointing this out.

**Minor/Specific comments:**

- Please avoid the wording "departure", i.e., neg. or pos. departures, it is not obvious what is meant with that phrase.

Thank you for your comment. We have replaced the word "departure" throughout the manuscript and made the necessary adjustments. In many cases, we have opted for the term "differences," which is more commonly used in remote sensing literature, to enhance clarity.

- Line 58: "…affects the atmospheric composition and radiative balance of the Middle East and western parts of Asia" – I guess the Eastern Mediterranean is missing here as well?

Thank you for pointing that out. We have updated the sentence to include the Eastern Mediterranean region as follows: 'The transport of dust from this region significantly affects the atmospheric composition and radiative balance of the Middle East, western parts of Asia, and the Eastern Mediterranean (Jish Prakash et al., 2015).

- Line 67: The references given for "passive and active remote sensing techniques" seem not to be appropriate especially with respect to the active instruments. Later on, the authors cite relevant publications with respect to active remote sensing - such should be also used here

Thank you for pointing this out. We agree that the original references were more focused on passive remote sensing techniques. In response to your suggestion, we have revised the sentence and updated the reference list to include key studies related to *active* remote sensing techniques, particularly lidar-based observations. The revised text now cites: Groß et al., 2011; 2015; Burton et al., 2012; Baars, et al., 2016; Benkhalifa et al., 2017; Haarig et al., 2017a; 2017b; 2022; Ningombam et al. 2019; Hofer et al., 2020; Raptis et al. 2020; Yu et al., 2022. These additions better reflect the use of ground-based passive and active techniques towards deriving long-term aerosol intensive and extensive properties and address the comment appropriately.

- Line 77: Cannot find reference Hansen 1995, Mishenko 1997 in the list. Thus, the connection to MAP is not given.

Thank you for pointing this out. We have added the references to Hansen et al. (1995) and Mishchenko and Travis (1997) to the reference list. The connection to the multi-angle, multi-spectral polarimetric (MAP) measurements is now properly cited, and we have ensured all relevant references are included.

- Line 94: "Past studies indicated…" Please give references for past studies.

Thank you for your comment. We have updated the sentence to clarify that the reference is specifically to the study by Schuster et al. (2012). The revised sentence now reads: 'Schuster et al. (2012) indicated that CALIPSO's AOD is often underestimated by approximately 13% when compared to AERONET. This underestimation is attributed to both aerosol misclassification and inaccuracies in the modeled microphysics for certain aerosol types, such as polluted dust or smoke.'

- Line 109-111: "The adaptation of a more "realistic" dust LR, along with more reliable elastic lidar-derived AOD values, could contribute to a more robust aerosol typing." It is not clear to me why this should be the case – please clarify.

We thank the Referee for pointing this out. We agree that the original sentence was not sufficiently clear. Our intention was not to suggest that the lidar-derived AOD directly contributes to aerosol classification. Rather, we meant that the adoption of a more realistic dust lidar ratio primarily improves the retrieval of more accurate elastic lidar-derived AOD values. We have revised the sentence accordingly to avoid confusion.

**Revised sentence (Lines 109–111):**

"The adaptation of a more 'realistic' dust LR can enhance the accuracy of elastic lidar-derived AOD estimates."

- Line 138-139: "...lidar system with horizontal averaging and vertical resolution variable based on both wavelength and altitude..." I do not understand this sentence, please rephrase it.

Thank you for your comment. We have rephrased the sentence for clarity. The updated sentence now reads:

"CALIOP operates as a dual-wavelength backscatter lidar system (532 and 1064 nm) and a single-wavelength polarization lidar system (532 nm). The horizontal averaging and the vertical resolution of the system vary depending on the altitude and wavelength (Winker et al., 2019)."

See Table 3 in Winker et al., (2019) for detailed information related to the horizontal and vertical resolution for the two different CALIOP channels.

- 157: LIVAS database: how can this be accessed?

Thank you for your comment. As also mentioned in our response to a major comment, LIVAS products are not publicly available through an open-access repository. However, they can be accessed upon request by contacting the data providers: Vassilis Amiridis (vamoir@noa.gr), Emmanouil Proestakis (proestakis@noa.gr), or Eleni Marinou (elmarinou@noa.gr). We have added this information to the Data Availability section of the manuscript."

- 226: Would be good to list the AERONET stations here already.

Thank you for the suggestion. We have now listed the AERONET stations used in this study directly in the relevant section of the manuscript. We added:

"The AERONET stations used in this study include Agoufou, Banizoumbou, Medenine-IRA, Zinder_Airport, Karachi, Masdar_Institute, Mezaira, Shagaya_Park, and Solar_Village (see Fig. 1b). These stations were selected based on specific criteria described in Sections 3.1 and 3.4."

- Fig. 1b: Please show the same frame as Fig. 1a otherwise it is confusing (same lat lon box)

We thank the Referee for this valuable comment. In the revised version of the manuscript, we have modified Fig. 1b to match exactly the same map extent and latitude/longitude ticks as Fig. 1a. This ensures a consistent frame between the two panels and improves the clarity and comparability of the figures.

**Fig.1b before modification:**

[Figure]

**Fig 1b after modification:**

[Figure]

**versus Fig. 1a**

[Figure]

- Figure 2 is quite blurry, make sure to have high resolution plots when resubmitting

We thank the Referee for pointing out the issue with Figure 2. This observation has been very helpful in improving the clarity of the figure. In response, we have recreated the plots in higher resolution and have also increased the fontsize and fontweight of the figure elements to enhance readability. We appreciate the feedback that has contributed to improving the quality of Figure 2.

- Caption Fig. 2. Please try to add the CALIOP aerosol subtype description in the plot.

We thank the reviewer for the valuable suggestion regarding the caption of Figure 2. We realize that the absence of titles for each panel may have led to some confusion. In response, we have added titles to each panel to clearly highlight the parameters represented in each plot. Additionally, we have increased the fontweight and fontsize of the aerosol subtype description at the bottom of Fig. 2b to make it more prominent and easily identifiable for the reader.

In summary, we have applied several modifications to Figure 2 compared to its previous version. These include the addition of titles for each panel to clarify the parameters represented, as well as an increase in the fontweight and fontsize of the aerosol subtype description at the bottom of Fig. 2b to make it more prominent. We hope these changes make the figure clearer and more informative for the reader.

**Fig 2 before modification:**

[Figure]

**Fig 2 after modification:**

**CALIPSO orbit: 2008-04-17T12-02-27ZD**

[Figure]

- Line 396: Why are the deserts in central Asia a blind spot and your approach cannot be used here?

Thank you for your comment. We apologize for any confusion caused. In the manuscript, when we referred to the regions over the Karakum and Kyzylkum deserts as a "blind spot," we did not mean to imply that our methodology cannot be applied to these areas. Rather, we were highlighting that these regions are relatively understudied in terms of dust lidar ratio (LR), with limited available observational data for comparison. The only study we found in the literature is by Hofer et al. (2020), which reports LR values in the range of 35-40 sr over Dushanbe, Tajikistan, based on measurements from the CADEX (Central Asian Dust EXperiment) campaign (Althausen et al., 2019). We have revised the manuscript to clarify this point and provide a more accurate context for the current state of research in this region.

- Line 396: Hofer et al 2020 not in reference list

Thank you for pointing this out. **Hofer et al., 2020** has now been added to the reference list. This correction is part of our broader revision addressing the 7th major comment, where we carefully reviewed and corrected the entire reference section to include all missing sources and fix typographical errors.

- Line 431: "summer months" should be avoided in this context for regions close to the equator

Thank you for this valuable comment. We acknowledge that the use of terms such as "summer" and "winter" can be ambiguous, particularly in regions near the equator. To avoid confusion, we have replaced these expressions with the corresponding meteorological season names — "June–August (JJA)" and "December–February (DJF)" — throughout the manuscript for greater precision.

- Figure 5: the distributions of the "LR boxes" seem a bit more extended (especially to the east) compared to the areas in Fig 1a, please comment!

Thank you for your comment. You are correct — the "LR boxes" in **Figure 5** appear more extended, particularly toward the east. While **Figure 1a** presents the predefined 2°×5° latitude–longitude grids used for computing the lidar ratio (LR), the Middle East domain from Ginoux et al. (2012) was intentionally extended slightly eastward, but not to the extent of overlapping with East Asian deserts. This adjustment was made to ensure that the region of focus, including the Arabian Desert, parts of Central Asia, and the western Indian subcontinent, was adequately covered. LR values were computed from CALIPSO overpasses that intersect both the 2°×5° grids and the adjusted Middle East regions (magenta colored), which explains the observed extension. To clarify this, you can see below both figures for comparison.

**Figure 1a:**

[Figure]

**Figure 5:**

[Figure]

- Line 473: "(via the integration of the vertical dust extinction coefficient profile)" here you could nicely refer to an equation once introduced in the methodology part.

Thank you for your valuable comment. In response, we have updated the methodology section to include the relevant equations, as suggested, particularly for the retrieval of DOD from CALIOP. We have now added the equation number to the section in line with the modifications made earlier, ensuring a clearer reference to the formula as part of the revised explanation.

We trust this change provides the necessary clarity, especially for non-lidar experts.

- Line 490. I propose to make Section 4.4 a new chapter 5. It is a completely different topic and after the "validation" part the respective "application" part. Thus, it would fit better to have a dedicated chapter.

We agree with the comment raised by the Reviewer and we have followed his/her recommendation in the revised manuscript.

- Line 494-496: I do not understand this: Previously it seems that you just use pure dust cases. Now something different is stated. If a new approach is used here, it should have been explained in the methodology section already. Can you explain what is the difference here compared to the sections above?

We thank the Reviewer for the comment. In Section 3.1, related to the selection of "pure" dust cases, we did not implement the methodology (discrimination technique) used to separate dust from dusty mixtures. This has been followed based on the fact that the (very) strict applied criteria effectively eliminate "contamination" by non-dust species, as also indicated by the very high particulate depolarization values. However, in Section 5.1, which presents the climatological DOD maps (based on the default and the updated dust LRs), the entire dataset of the CALIPSO profiles is processed without applying the criteria described in Section 3.1. Consequently, we must manage a very large number of CALIPSO profiles without being able to "constrain" each one individually. Since we are focusing on the assessment of the revised dust LRs to climatological DODs, we are applying the discrimination technique to account for dust extinction in cases where other aerosol types contribute to the aerosol burden.

- Line 639: "In elastic lidars (such as CALIOP), for the derivation of the aerosol extinction coefficient from the backscatter coefficient, it is required a factor known as lidar ratio". Actually, it is also needed to retrieve the backscatter coefficient - see major comment 3.

Thank you for the clarification. We agree with the Reviewer that our initial statement was misleading and we have revised this sentence accordingly.

"In elastic lidars (such as CALIOP), for the derivation of the aerosol extinction and backscatter coefficients it is required a factor known as lidar ratio"

- In the conclusion section, it would be good to emphasize that higher LR mean higher DOD and vice versa.

We thank the reviewer for this insightful suggestion. In the revised manuscript, we have enhanced the conclusion section by explicitly stating the physical relationship between the lidar ratio (LR) and dust optical depth (DOD). While the original version of the conclusion discussed the spatial variability and impact of the adjusted LR values on the resulting DOD fields, it did not clearly explain the underlying dependency between the two quantities. To address this, we have inserted the following clarification:

"Since the DOD is directly proportional to the lidar ratio, regions with higher LR values result in enhanced DODs, whereas lower LRs yield reduced DODs."

This addition not only responds directly to the reviewer's request but also improves the clarity of our concluding arguments by reinforcing the interpretation of our results. It makes the link between the retrieved LR values and the observed DOD changes more accessible, especially to readers less familiar with lidars.

- Line 647: "…which are less pronounced in the CALIOP-POLDER results." Why?

We thank the Referee for this comment. We have revised the sentence to better clarify this point, while acknowledging the limitation in attributing the exact cause of the observed differences. The revised sentence reads:

"The CALIOP-MIDAS synergy reveals higher lidar ratios over central Asia—particularly over the Karakum and Kyzylkum deserts—which are less pronounced in the CALIOP-POLDER results, possibly due to differences in sensor characteristics, retrieval algorithms, or temporal coverage."

While we cannot pinpoint a single reason, the differences likely stem from the fundamentally distinct nature of the two datasets. MIDAS is based on MODIS-Aqua AOD retrievals combined with MERRA-2 reanalysis to isolate the dust component, whereas POLDER-3 relies on polarimetric and multi-angle observations processed through the GRASP/Component algorithm to retrieve aerosol properties. These methodological differences affect the representation of aerosol types and their optical properties.

Additionally, the two datasets cover different time periods: MIDAS spans 2007–2017, while the POLDER-3/GRASP product used here is limited to 2006–2009. Differences in interannual variability of dust emissions, transport, or meteorological conditions could also contribute to the spatial patterns observed in the respective lidar ratio estimates.

More details in respect to MIDAS and POLDER-3/GRASP can be found in Sections 2.2 and 2.3.

- Line 653: "each synergy (combination) performing better or worse" Do they perform better or worse? Both is not possible.

And

- Line 655: "showed a notable decrease or increase" → notable deviation

Thank you for your helpful suggestions. We agree that the phrase "each synergy (combination) performing better or worse" was unclear and may have given the impression that a single synergy simultaneously performed both better and worse, which was not our intention.

What we meant to convey is that the performance of each synergy varied depending on the station — in some cases, one synergy produced better results, while in others, a different synergy was more accurate. To clarify this, we have revised the sentence accordingly.

Additionally, we appreciate the suggestion to replace "showed a notable decrease or increase" with the more concise and precise phrase "notable deviation." This improves the readability and avoids redundancy.

The revised text now reads:

"In particular, the best improvements were recorded over Karachi, Masdar Institute, and Mezaira, with the performance of each synergy (combination) varying across stations — in some cases one synergy performed better, while in others, another was more accurate. The deviations from the DOD based on the CALIPSO default LR were more significant at stations where the LR showed a notable deviation from the default value of 44 sr."

We believe this revision better reflects our intended meaning and addresses both comments.

- 691: Please make LIVAS available as well as your research data set.

We fully agree with the importance of open data sharing for scientific transparency and reproducibility. However, as previously stated, the LIVAS dataset is not developed or maintained by us, and we do not hold the rights to redistribute it publicly. For this reason, we cannot upload it to a public repository. Nevertheless, as indicated in the Data Availability section, the dataset is available upon request from the official data providers: Vassilis Amiridis (vamoir@noa.gr), Emmanouil Proestakis (proestakis@noa.gr), and Eleni Marinou (elmarinou@noa.gr).

Regarding the novel CALIPSO dust extinction dataset produced in this study, we will provide a NetCDF file containing the derived lidar ratio (LR) values across North Africa and the Middle East from both synergies (CALIOP–POLDER-3/GRASP and CALIOP–MIDAS). This dataset will be uploaded to a

publicly accessible repository (e.g., Zenodo), and the relevant link will be included in the final Data Availability section.

We hope this approach addresses the comment and fulfills the expectations regarding data accessibility.

- Line 706: I think it would be fair to also acknowledge the PI's of the respective AERONET stations by name.

Thank you for your valuable suggestion. In response, we have updated the Acknowledgment section of the manuscript to specifically acknowledge the Principal Investigators (PIs) and Co-Investigators of the AERONET stations used in this study. The revised wording now reads:

"We are grateful to AERONET for providing high-quality sun-photometer observations, calibrations, and products. Special thanks to the Principal Investigators and Co-Investigators of the AERONET stations used in this study: Philippe Goloub (Agoufou), Didier Tanré, Jean-Louis Rajot (Banizoumbou), Gilles Bergametti (Medenine-IRA), Jean-Louis Rajot, Mahammadou M. Zakari (Zinder Airport), Pawan Gupta, Elena Lind, Muhammad Ateeq Quershi (Karachi), Matteo Cheisa (Masdar Institute), Pawan Gupta, Elena Lind (Mezaira & Solar Village), and Alaa Ismail (Shagaya Park).

We believe this revision now appropriately acknowledges the contributions of the relevant PIs and Co-Investigators. Thank you again for this helpful suggestion.